

# Influence of groundwater recharge projections on climate-driven subsurface warming: insights from numerical modeling

Mikhail Tsypin[1,2], Viet Dung Nguyen[1], Mauro Cacace[1], Guido Blöcher[1,2], Magdalena Scheck-Wenderoth[1,2], Elco Luijendijk[3], Charlotte Krawczyk[1,2]

[1] GFZ Helmholtz Centre for Geosciences, Potsdam, 14467, Germany
[2] Technische Universität Berlin, Berlin, 10623, Germany
[3] University of Bergen, Bergen, 5007, Norway

*Correspondence to:* Mikhail Tsypin (tsypin@gfz.de)

**Abstract.** Groundwater warming due to rising surface temperatures has been documented in both urban and
natural settings. However, the potential for long-term changes in the magnitude and seasonality of groundwater
recharge to modulate this warming trend has not yet been systematically investigated. In this study, we integrate
a stochastic weather generator, distributed hydrologic modeling, and regional thermo-hydraulic groundwater
modeling into a unified workflow and apply it to the area of Brandenburg (northeastern Germany). We conduct
numerical simulations to assess changes in the subsurface thermal field between present day and 2100, evaluating
two climate change scenarios, and incorporate a spectrum of ensemble-based and discrete recharge projections.
Our results demonstrate that, while surface temperature rise is the primary driver of the projected groundwater
warming of up to 2.5 °C, groundwater flow is responsible for its regional variability in magnitude and affected
depths. Higher hydraulic gradients on topographic highs and increased thickness of the permeable Quaternary unit
may allow the warming signal to propagate below 200 m depth, whereas groundwater discharge in the river valleys
tends to limit it to <200 m. By the late century, the difference in groundwater temperatures between recharge-
reduction and recharge-increase scenarios can reach 0.4 °C. Under the high-emissions pathway, a 20 % recharge
reduction, from a mean of 75 to 60 mm a⁻¹, causes a 2–5 m water level decline, reducing the area of unconfined
aquifer subjected to seasonal temperature fluctuations. Model experiments show that even a hypothetical increase
in winter recharge does not suffice to counteract the groundwater warming induced by rising surface temperatures.
Changes in advection rates are not expected to affect the net climate-driven accumulated heat in the subsurface
due to counterbalancing of heat gains and losses between recharge and discharge areas. Nevertheless, long-term
reconfiguration of the potentiometric surface may further impact both the annual and long-term thermal state of
key aquifers targeted for water supply and shallow geothermal energy utilization.

## 1   Introduction

The ongoing climate change modifies components of the Earth's water and energy balance, thereby affecting the
hydraulic, thermal, chemical, and biological states of groundwater systems (Taylor et al., 2013; Riedel, 2019). In
line with other climate change impact assessment studies (Seneviratne et al., 2010; Gosling et al., 2011), predictive
modeling of these groundwater states requires a consistent set of climate scenarios, ideally based on ensembles of
general circulation models (GCMs) and $CO_2$ emission pathways. Among GCM outputs, temperature and
precipitation are considered key variables controlling surface-groundwater fluxes, such as infiltration,
evapotranspiration, and recharge (Moeck et al., 2016; Rossman et al., 2014).




On global and regional scales, groundwater availability for human consumption and irrigation strongly depends on recharge rates (Cuthbert et al., 2019; Reinecke et al., 2021). A prediction of future groundwater levels (GWLs) using analytical, numerical, and machine learning tools has been widely explored across various timescales, being

increasingly adopted by water management authorities for planning purposes (Scibek and Allen, 2006; Dams et al., 2012; Goderniaux et al., 2011; Seidenfaden et al., 2022; Wunsch et al., 2022). A major uncertainty though still exists in future recharge estimates, as there is a lack of consistency across models concerning both the magnitude and sign of recharge trends (Smerdon et al., 2017). This stems largely from uncertainty in intensity and distribution of precipitation rather than potential evapotranspiration (PET) due to the dependence of the former

on circulation patterns and higher spatiotemporal variability (Reinecke et al., 2021; Berghuijs et al., 2024). Recharge estimates are therefore sensitive to the choice of GCM, a Shared Socioeconomic Pathway (SSP), the downscaling approach, and hydrologic model parameters (Kurylyk et al., 2015).

More recently, the thermal response of aquifers to climate change came into the spotlight, following reports of groundwater warming in urban centers (Tissen et al., 2019; Menberg et al., 2013) and due to the growing

utilization of shallow geothermal energy (Hemmerle et al., 2022). Implications of a sustained increase in groundwater temperatures, induced by climate change and urbanization, include the potential for shallow subsurface heat recycling and impacts on temperature-sensitive aquifer and stream ecosystems and groundwater quality (Benz et al., 2024; Hemmerle and Bayer, 2020; Riedel, 2019; Kurylyk et al., 2015).

Subsurface temperatures are projected to rise due to the concomitant rise in surface temperatures, which are

expected to reach 1.5 °C to 5 °C above the pre-industrial levels by the end of the century, depending on the emission scenario (IPCC, 2023). However, the additional effect of changes in groundwater recharge in modifying heat advection by groundwater flow has received less attention to date (Klepikova, 2024), being mostly described by analytical 1-D models (Taniguchi et al., 1999; Kurylyk et al., 2017) or conceptual 2-D numerical models (Benz et al., 2024; Zhang et al., 2024; Irvine et al., 2016).

To our knowledge, there is so far no consensus on whether projected recharge trends can offset, if not counterbalance the conductive rise of groundwater temperatures. A study by Benz et al. (2024) stated that downward propagation of the surface warming signal caused by climate change is primarily driven by thermal diffusion, with changes in advection rates due to groundwater recharge playing a minor role. Other authors arrived at contrasting conclusions, arguing that projected changes in precipitation amount and seasonality, and thus

recharge temperature and/or groundwater fluxes, could alter the groundwater warming trend from rising surface temperatures (Epting et al., 2021; Burns et al., 2017; Kurylyk et al., 2013).

The modification of the conductive geothermal gradient by advection has been used to quantify vertical groundwater fluxes from temperature profiles (Stallman, 1965; Bredehoeft and Papaopulos, 1965; Anderson, 2005). Likewise, an inversion of borehole temperature logs may provide information about past land surface

temperatures for paleoclimatic studies, though this normally assumes only conductive heat transport (Shen and Beck, 1992). It remains uncertain whether changes in groundwater recharge of up to $\pm100$ mm a$^{-1}$, projected globally by the end of the century (Reinecke et al., 2021), are sufficient to alter aquifer thermal responses to surface temperature forcing. A major challenge in assessing recharge effects lies in the influence of preferential flow in the unsaturated zone and the predominance of the lateral flow in the saturated zone due to topographic

gradients.



Topography-driven groundwater flow exerts a footprint on the subsurface thermal field, leading to decreased heat flow in recharge areas and increased heat flow in discharge areas, an aspect that has been demonstrated in basin- and lithosphere-scale thermo-hydraulic models (Kooi, 2016; Noack et al., 2013; Tsypin et al., 2024; Smith and Chapman, 1983). However, field evidence of regional temperature variability from advective groundwater flow has been limited to favorable settings for deep meteoric water circulation, such as mountainous terrains and rift basins (Deming et al., 1992; Luijendijk, 2012). The extent of groundwater cooling depends on both hydraulic gradients and rock permeability, though some authors argued that rates of groundwater flow are often too small to perturb the regional thermal field (Bachu, 1988).

The objective of the present study is to investigate the coupled effects of rising surface temperatures and temporal trends of groundwater recharge on the subsurface thermal field from the present day to the end of the century when considering regional controls on heat advection, such as multidimensional groundwater flow and geological heterogeneities.

To achieve this objective, we employ a non-stationary weather generator to produce an ensemble of downscaled climate projections, corresponding to different GCM-SSP combinations. The weather forcing is used to calculate groundwater recharge fluxes via a hydrological model, that are assigned, along with projected surface temperatures, as time-variable boundary conditions to a regional 3-D groundwater model. We then conduct a series of transient thermo-hydraulic simulations to assess long-term changes in hydraulic heads, groundwater velocities, and heat in place, and discuss potential implications for groundwater storage and geothermal energy availability.

We apply our modeling approach to the study area of Brandenburg, NE Germany, where a periglacial geology and geomorphology result in complex deep-to-shallow groundwater dynamics, and where ongoing climate change has already increased stress on water and energy resources (Alencar et al., 2025; Brill et al., 2024). The presented model, while being limited in covered area and timespan, provides an opportunity to assess the role of the individual co-players (temperate climate, recharge uncertainty, permeable shallow section, undulating water table) affecting the groundwater system response under long-term climatic changes.

## 2    Study area

The study area spans the territory of Germany's Federal State of Brandenburg and the city of Berlin, as representative of a Central European climate and geology (Fig. 1a).

The region has a temperate continental climate, with a mean annual temperature of +9.0 °C, January being the coldest month and July the warmest. Mean precipitation is 550 mm a$^{-1}$, February being the driest and June the wettest month (GERICS - Climate-Service Center Germany, 2019).

Geologically, the study area delimits the southeastern margin of the intracontinental North German Basin (NGB). The basin fill is composed of Permo-Cenozoic sedimentary and volcanic rocks, with total thickness increasing from <100 m on uplifted basement highs in the south to >8 km in the basin center northwest of the study area (Littke et al., 2008; Krawczyk and Schulze, 2015). The deep lithospheric structure, basement composition, and sedimentary cover blanketing contribute to an average surface heat flow of 65–75 mW m$^{-2}$ (Fuchs et al., 2022). Local heat flow variations are influenced by Upper Permian Zechstein evaporites, which form diapirs and salt walls, acting as vertical conduits for heat. The Middle Triassic Muschelkalk formation, composed of marine carbonates and mudstones, is often viewed as an aquitard in basin-scale modeling (Frick et al., 2022a).





Consequently, our modeling effort focuses on the overlying Upper Triassic to Quaternary portion of the basin, characterized by the presence of permeable and locally interconnected reservoirs, and hence a more active groundwater circulation.

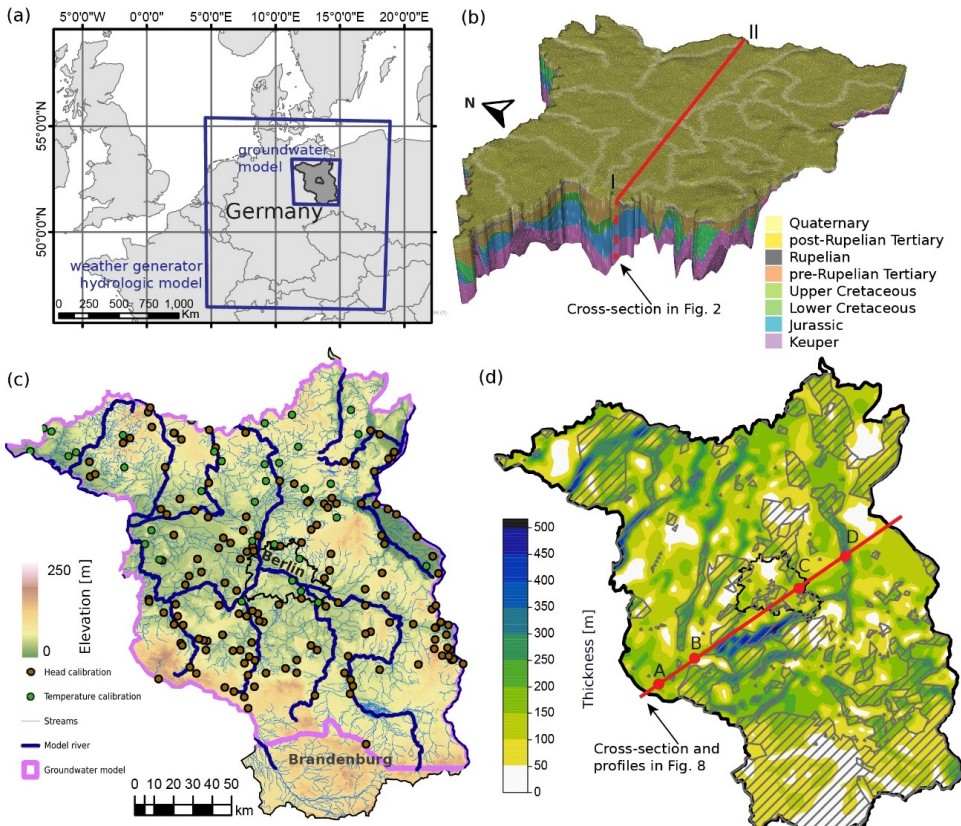

**Figure 1: Overview of the study area: (a) geographical location and models' extent; (b) 3-D view of the groundwater model mesh; (c) topography of Berlin-Brandenburg and locations of the groundwater model calibration points; (d) thickness of the Quaternary unit. Dashed polygons indicate absence of the Rupelian aquitard.**

Keuper (uppermost Triassic), Jurassic, and Lower to Upper Cretaceous deposits consist of interbedded mudstones

and sandstones, and limestones, with a total thickness of up to 3 km (Stackebrandt and Manhenke, 2010). These saline aquifers are utilized for low-enthalpy geothermal resource extraction and for heat and gas storage (Bruhn, 2023).

Tertiary deposits include a mix of shallow marine, nearshore, and continental clastic sediments, including coal deposits. Within the Tertiary section, the Rupelian Clay is a key marine transgressive layer, acting as a basin-wide

aquitard separating predominantly freshwater aquifers above from brackish to saline aquifers below (Stackebrandt and Franke, 2015).

Quaternary sediments consist of poorly consolidated fluvioglacial and alluvial deposits: sand, silt, clay, and gravel. The retreat of pressurized subglacial meltwater during the Pleistocene led to the formation of SW-NE-oriented



tunnel valleys, incised up to 550 m deep, locally eroding the Rupelian Clay (Lang et al., 2025) (Fig. 1d). These
valleys are filled with coarse-grained clastic deposits that allow for focused cross-flows between deep and shallow
aquifers, locally leading to freshwater salinization. The heterogeneous architecture of Quaternary glacial deposits
has produced a complex hydrogeological system, with several hydrodynamically interconnected aquifer
complexes. Quaternary aquifer complexes are used for drinking water supply and are increasingly targeted by
ground-source heat pumps (GSHP) and low-temperature aquifer thermal energy storage (LT-ATES) systems
(Ministerium für Wirtschaft, 2023; Stemmle et al., 2025).

The present-day relief, shaped during Pleistocene glaciations, consists of alternating glacial plateaus and ice-
marginal valleys, hosting major rivers (Sirocko et al., 2008). Absolute elevations range between 50–200 m a.s.l.
on glacial plateaus to 0–30 m a.s.l. in river valleys (Fig. 1c). The depth to the water table varies from 0–10 m in
floodplains and river valleys to up to 50 m on glacial plateaus. Soils are dominated by sands and loamy sands with
low water-holding capacity (Stackebrandt and Franke, 2015).

### 3    Methods

#### 3.1    Simulation of downscaled weather time-series

We employed a stochastic weather generator to derive ensemble-based, spatially and temporally consistent
downscaled precipitation and temperature inputs for the subsequent hydrologic and groundwater modeling.
Weather generators have been widely used to simulate probabilities of extreme hydrologic events such as droughts
and heavy precipitation and are also valuable to model the stochastic component of climatic variability in
groundwater resource studies (Goderniaux et al., 2011). In our study, we rely on the non-stationary Regional
Weather Generator (nsRWG; Nguyen et al. (2024)). This tool simulates daily weather sequences that capture both
natural variability and evolving climate conditions, making it well-suited for long-term hydrological assessment
studies. nsRWG combines a multivariate auto-regressive spatio-temporal structure for the weather variables of
interest with marginal distributions conditioned on large-scale climate drivers. These drivers include circulation
patterns (CPs) derived from mean sea level pressure and regional daily mean temperatures, capturing both
synoptic-scale variability and thermodynamic trends.

The model was configured for a Germany-wide domain realization, covering 540 grid cells, using the gridded
observational dataset E-OBS version 25.0e (Cornes et al., 2018). This dataset provides daily precipitation and
minimum, maximum, and mean near-surface air temperatures for the period 1950–2021. nsRWG has been
rigorously evaluated and has demonstrated good performance in reproducing site-specific and spatial
characteristics of precipitation and temperature (Nguyen et al., 2024).

In this study, we use nsRWG to simulate long-term daily weather series for both the historical (1950–2021) and
future (2022–2100) periods. Historical simulations are driven by CPs and regional temperatures derived from the
ERA5 reanalysis dataset (Hersbach et al., 2020). Future simulations are based on projections of CPs and regional
temperatures from seven GCMs of the IPCC Coupled Model Intercomparison Phase 6 (CMIP6): CNRM-CM6-1,
INM-CM5-0, GFDL-ESM4, MPI-ESM1-2-HR, MRI-ESM2-0, CESM2, and IPSL-CM6A-LR, see also Nguyen
et al. (2024). We consider two climate scenarios: SSP245, a moderate-emission pathway where mitigation efforts
balance sustainability measures with economic growth, and SSP585, a high-end emission scenario driven by fossil



fuel development with minimal mitigation (IPCC, 2023). We generate 50 stochastic realizations both for the historical case and for each GCM–SSP combination, resulting in a total of 750 realizations.

## 3.2 Recharge estimation

Meteorological forcing generated by nsRWG served as input to the mesoscale Hydrologic Model (mHM) (Samaniego et al., 2010; Samaniego et al., 2019), which is used to simulate groundwater recharge. mHM is a spatially distributed, process-based model that approximates hydrologic fluxes such as soil moisture dynamics, infiltration, surface runoff, evapotranspiration, groundwater recharge, discharge generation, and baseflow.

Conceptually, mHM includes three subsurface storage compartments: a multi-layer soil reservoir, an unsaturated subsurface reservoir, and a groundwater reservoir. Fluxes between these compartments are computed locally at each grid cell, while runoff routing follows the steepest topographic gradient.

mHM applies a Multiscale Parameter Regionalization (MPR) technique (Kumar et al., 2013), which considers morphological variables as predictors to derive spatially coherent parameter fields. This approach enables the model to produce seamless and physically consistent outputs across regions with diverse hydrological behavior.

The model was set up over a domain encompassing Germany and upstream contributing areas in neighboring countries (Fig. 1a). The setup relies on two primary datasets: (1) meteorological forcing and (2) physiographic input. The model was forced with daily precipitation, near-surface air temperature, and PET. The latter was estimated using the Samani–Hargreaves method based on daily minimum, mean, and maximum temperatures (Hargreaves and Samani, 1985). Physiographic inputs comprised a digital elevation model, along with maps detailing soil properties, land use, land cover, and reservoir lithology. Meteorological forcings were organized at a spatial resolution of 25 km, whereas the physiographic inputs were prepared at a finer resolution of 5 km.

Model calibration was conducted using a regional approach based on MPR and validated against streamflow data from 102 hydrological gauges across Germany (Guse et al., 2024). Groundwater recharge have been additionally validated against GWL in a transient groundwater model of Brandenburg (Tsypin et al., 2025).

## 3.3 Groundwater flow and heat transport modeling

We conducted coupled groundwater flow and heat transport simulations using FEFLOW 8.0 (Diersch, 2013) and PEST calibration package (Doherty, 2015). Groundwater flow was solved according to the equation:

$$S_s \frac{\partial h}{\partial t} + \nabla \cdot q = R, (1)$$

where $S_s$ is specific storage [m$^{-1}$], $h$ – hydraulic head [m], $t$ – time [s], $R$ –volumetric source/sink term [m$^{-1}$] (e.g., groundwater recharge), and $q$ – groundwater flux [m s$^{-1}$], which, according to Darcy's law, is defined as:

$$q = -K\nabla h, (2)$$

where $K$ is the hydraulic conductivity [m s$^{-1}$].

3-D heat transport equation accounted for changes in thermal storage, advective and conductive heat exchange, and radiogenic heat production:

$$(\rho c)_b \frac{\partial T}{\partial t} - \nabla \cdot \left(\lambda_b \nabla T - (\rho c)_f q T\right) - Q_r = 0, \quad (3)$$

where $\rho$ – density [kg m$^{-3}$], $c$ – specific heat capacity [J kg$^{-1}$ K$^{-1}$], $b$ and $f$ subscripts refer to bulk and fluid properties, respectively, $T$ – temperature [K], $\lambda$ – thermal conductivity [W m$^{-1}$ K$^{-1}$], and $Q_r$ –radiogenic heat production [W m$^{-3}$].





Dispersion effects and free convection were neglected for the heat transport, given the regional scale of the model with a focus on shallow groundwater circulation, and fluid density and viscosity were assumed constant.

The finite-element mesh incorporated structural surfaces from the regional geological model of Brandenburg (Noack et al., 2024). The generated mesh is 170×150 km wide and up to 2.5 km deep, consisting of 8 stratigraphic units, described in Sect. 2. The Quaternary unit was subdivided into 10 simulation layers, and post-Rupelian Tertiary and Rupelian units were split into 3 layers each, proportional to each unit's total vertical thickness. The mesh was further refined along 19 major streams to improve computational stability and accuracy in simulating

the shallow subsurface thermal and hydraulic behavior (Fig. 1b). The final mesh consists of 1.15 million triangular prismatic elements and 615,000 nodes.

Thermal and hydraulic properties for stratigraphic units were kept identical across scenarios to isolate climatic controls (Table A1) and were taken from previous studies (Scheck and Bayer, 1999; Noack et al., 2013) except Quaternary and Tertiary post-Rupelian units, which were subject to calibration.


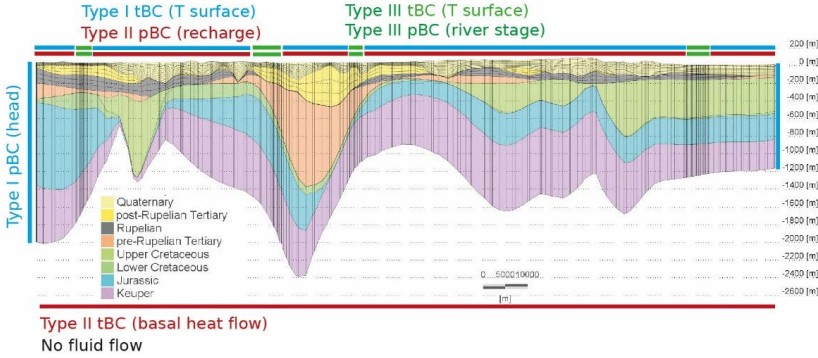

**Figure 2: Layering and the setup of the hydraulic (pBC) and thermal (tBC) boundary conditions of the groundwater model. The location of the cross-section is shown in Fig. 1b.**

The base of the model was taken as a no-flow boundary, corresponding to the Middle Triassic Muschelkalk formation. At lateral model edges a constant-head Type I (Dirichlet) boundary condition (BC) was assigned, given that they largely correspond to major rivers. At the top, a Type II (Neumann) BC was specified, using recharge rates derived from mHM, except for cells corresponding to river elements, where we applied a Type III (Cauchy) BC, with a time-invariant reference elevation across all scenarios (Fig. 2). The choice of this BC stems from the

fact that most streams in the study area have regulated levels with limited sensitivity to recharge fluctuations.

We assigned a spatially variable basal heat flux as the lower thermal BC, derived from a lithosphere-scale thermal model of Brandenburg, ranging 55–95 mW m$^{-2}$ (Noack et al., 2012). Surface temperature was assigned as Type I BC or as Type III BC along the river nodes to allow for variable discharge temperature. To maintain consistency between the hydrologic and groundwater models, we used near-surface air temperature as a proxy for land surface

temperature. An analysis of monthly differences between near-surface air and land surface temperatures across CMIP6 climate models for the study area revealed that in 99 % of cases the discrepancy was <0.4 °C.



The unsaturated zone was approximated via a phreatic surface approach (Diersch, 2013; Desai and Li, 1983). Saturated hydraulic conductivity in partially saturated elements was linearly scaled by saturation, with a minimum residual saturation threshold of 1 %.

As initial conditions to the transient groundwater model, we selected the pressure-temperature (PT) state in 1955, given that the difference in the measured air temperature between the onset of industrialization in the 1880s and 1950s is <0.3 °C in the study area (DeutscherWetterDienst - Climate Data Center, 2021). To derive initial heads and temperature conditions for the transient model, we assigned a 5-year mean surface temperature and recharge from the earliest available historical period and ran the model until steady state.

The transient model was calculated for 1955-2100 with adaptive time steps not exceeding monthly increments. Post-processing focused on comparing model outputs between two 20-year-long intervals: a control **present period** (2002-2021) and a **late-century period** (2081-2100). The model only considers natural climatic (exogenic) forcing, without including effects of urban climate/infrastructure or groundwater abstraction/irrigation. Therefore, the magnitude of simulated PT effects primarily represents gross regional trends.

The relative contributions of advection and diffusion to the heat transport were quantified via the dimensionless Peclet number (*Pe*):

$$Pe = \frac{vL}{D}, (4)$$

where $v$ is flow velocity [m s$^{-1}$], $L$ – characteristic length [m], determined as the size of the element in the flow direction, and $D$ is thermal diffusivity [m$^2$ s$^{-1}$].

We compute the incremental accumulated subsurface heat-in-place as:

$$Q_{\text{total}} = \sum_{i=1}^{N} V_i \left( \phi_i C_w + (1 - \phi_i) C_s \right) \left( T_{i,t2} - T_{i,t1} \right), (5)$$

with $V_i$ being the volume of the $i$-th element [m$^3$], $\phi_i$ the porosity of the $i$-th element, C the volumetric heat capacity of water (w) and solid (s) [J m$^{-3}$·K$^{-1}$], $T_{i,t2}$ and $T_{i,t1}$ the mean temperatures in the respective time periods, and $N$ the total number of finite elements in the model domain.

**4    Scenario Definition**

**4.1    Ensemble scenarios**

For the two climate scenarios (SSP245 and SSP585), we constructed ensembles of temperature, precipitation, PET, and groundwater recharge time series. Each ensemble combined seven GCMs with 50 realizations per GCM. For the historical period, the ensemble likewise consisted of 50 realizations. We then considered monthly mean

values for each ensemble for the entire Brandenburg area, as the modeled forcing showed only limited spatial variability in their long-term trends (Fig. A1 and A2).

The historical increase in mean annual temperature since 1950 (excluding the city of Berlin) is 1.1 °C. For SSP585, surface temperature in Brandenburg is projected to rise by an additional 3 °C on average by 2100 (P90 (ensemble's 90[th] percentile value): 3.5 °C, P10 (10[th] percentile): 2.5 °C), while under SSP245 the projected mean

increase is 1.5 °C (P90: 2.0 °C, P10:1.2 °C) (Fig. 3a).

Precipitation, despite having high variability driven by the stochastic component, does not exhibit a statistically significant long-term trend in the annual mean. Projections differ between the two scenarios only after 2075, with SSP585 portraying slightly lower annual precipitation.





PET time series closely follow temperature variations and, given relatively steady precipitation, point to a long-term decrease in the climatic water balance and an increase in aridity.

Simulated recharge shows a decline beginning in the late 1970s with a mean trend of -4 mm a$^{-1}$ per decade. This lies at the lower end of the trend of -21 to -4 mm a$^{-1}$ per decade, estimated for Brandenburg, and is supported by falling GWL in the observation wells, decreasing stream discharge, and groundwater storage reductions based on satellite hydrogravimetry (Francke and Heistermann, 2025; Güntner, 2023). This negative recharge trend has been attributed to climate change and increasing leaf area index (LAI) (Francke and Heistermann, 2025). An analysis of groundwater trends in watersheds minimally affected by anthropogenic forces suggests hydraulic heads declining at rates of -20 to -80 mm a$^{-1}$, resulting in up to 2.5 m cumulative decline from mid-1970s to the present. Simulated future recharge exhibits a continued negative long-term trend under both climate scenarios. Given that projected precipitation remains relatively stable, the decline is attributed to increasing PET and actual evapotranspiration. The total reduction of recharge during the projection period for SSP585 and SSP245 reaches up to -25 to -30 % and -6 to -12 %, respectively, in agreement with predictions based on an empirical relationship linking groundwater recharge with precipitation to PET ratio (Berghuijs et al., 2024).

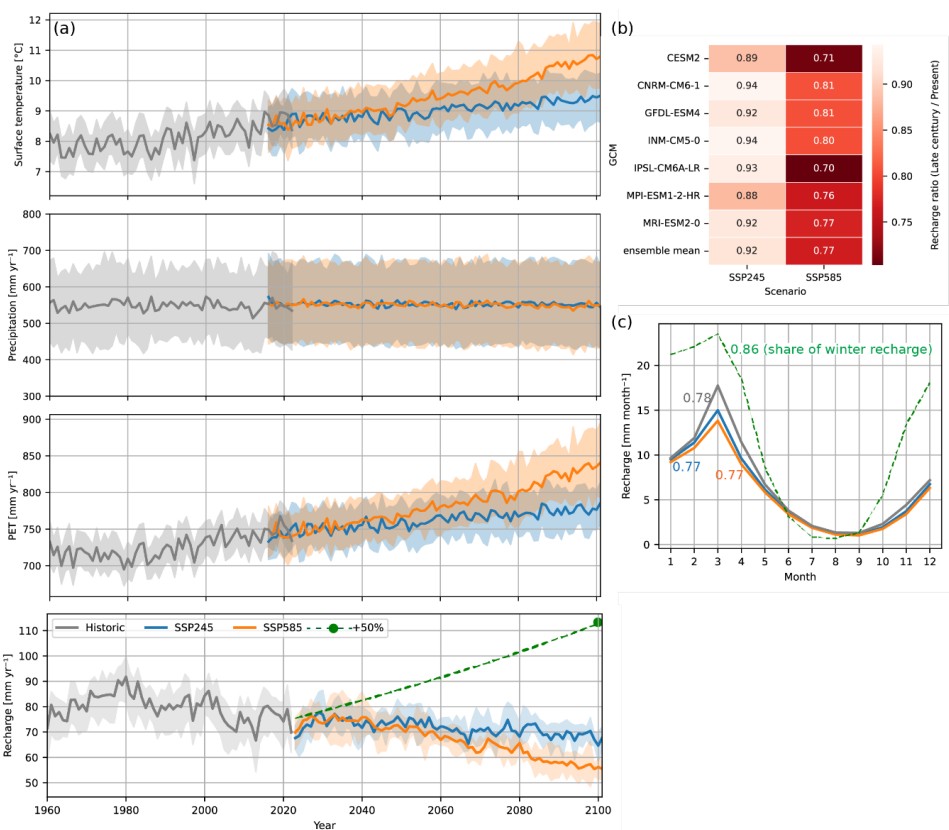

**Figure 3: (a) historical and projected annual time series of temperature, precipitation, PET, and groundwater recharge. Solid lines represent the ensemble mean (historical: across nsRWG realizations; SSP scenarios: across GCMs and realizations) and shaded areas indicate the P10-P90 range; (b) ratio of late-century to present period mean recharge; (c) monthly recharge means. The green line represents a discrete synthetic scenario (Sect. 4.2).**





Recharge projections show consistency across all GCMs (Fig. 3b, Fig. A2). CESM2 tends to give lower recharge
in both climate scenarios, while MPI-ESM1-2-HR and IPSL-CM6A provide the largest change only for SSP245
and SSP585, respectively. CNRM-CM6-1 exhibits the smallest change between the reference and the future
periods.

Simulated recharge also displays a strong seasonal signal, with >70 % of recharge occurring during the colder
months (November to April) for both historical and projected periods (Fig. 3c).

We calculated groundwater flow and heat transport using cross-GCM statistical ensembles (mean, P10 and P90
time series of the nsRWG suite) for each SSP, as well as six deterministic nsRWG realizations with the smallest
Root Mean Square Error (RMSE) relative to the aggregate scenarios. The ensemble approach captures the long-
term uncertainty range, while the deterministic cases retain temporal variability and extremes, helping to offset
potential artifacts from ensemble smoothing.

**4.2    Synthetic scenarios**

The use of the weather generator enabled us to capture recharge variability while conditioning on large-scale
circulation patterns (synoptic dynamics) and atmospheric thermodynamic effects. However, as demonstrated in
multi-model hydrological studies (Moeck et al., 2016; Reinecke et al., 2021; Kumar et al., 2025), uncertainties in
future recharge estimates arise not only from the choice of GCM, with their diverse precipitation projections, but
also from the hydrologic model setup (e.g., vegetation processes considered, PET estimation method) and its
dynamic inputs (e.g., LAI and land use change).

While the scope of the current study does not include a multi-model ensemble analysis, we examined available
alternative recharge projections for the region of interest (Kumar et al., 2025; Berghuijs et al., 2024; Hattermann
et al., 2008; Marx et al., 2021). Central Europe lies in a transitional climatic zone between the Mediterranean
region, where projections show a pronounced decrease in recharge, and Northern Europe/Scandinavia, where most
projections indicate an increase. To account for these contrasting regional trends in our groundwater simulations,
specifically the implications of increasing recharge despite rising PET, we included a scenario based on a projected
end-century increase of 23 % (overall for Germany) and up to 50 % (locally in Brandenburg) (Marx et al., 2021).
Despite concerns over such "optimistic" projections driven by increased winter precipitation (Tillman and
Francke, 2025), we found such a "what if" scenario valuable for assessing potential subsurface feedbacks. Taking
this projected increase, we generated synthetic monthly recharge time series assuming exponential growth over
time and a higher winter share of total recharge (Fig. 3). We tested the effect of increasing recharge with moderate
(Scenario **M1**, Table 1) and strong warming (**M2**). In order to isolate the flux effects, we also tested two cases
where we maintain surface temperature at their present values and consider a 20 % recharge decline (as in SSP585,
**M3**) or a hypothetical increase of 50 % (**M4**).

The resulting layout of scenarios for groundwater simulations allows us to compare the difference between two
climate scenarios and a potential impact of diverging recharge trends on the subsurface flow and temperature
distribution (Fig. 4).




**Table 1: Summary of the tested groundwater modeling scenarios.**

| Scenario | Description | ΔT (surface) | Δ Groundwater recharge | Comment |
|---|---|---|---|---|
| | | **Present to late-century periods** | | |
| | *Ensemble scenarios derived with nsRWG and mHM* | | | |
| SSP245 | Moderate emission pathway | +1.5 °C | -10 % | Ensemble mean |
| SSP585 | High-end emission pathway | +3 °C | -20 % | |
| | *Synthetic and reference scenarios* | | | |
| M1 | Recharge increase, moderate warming | +1.5 °C | +50 %, higher winter recharge ratio | after Marx et al. (2021) |
| M2 | Recharge increase, strong warming | +3 °C | +50 %, higher winter recharge ratio | |
| M3 | Recharge as SSP585, present-day temperature | 0 °C | -20 % | Reference cases to isolate the recharge effects |
| M4 | Recharge increase, present-day temperature | 0 °C | +50 % Winter | |

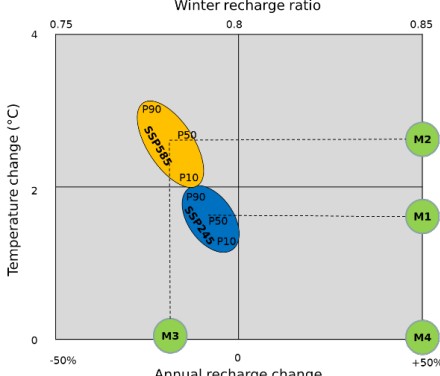

**Figure 4: Schematic representation of the groundwater modeling scenarios. See Table 1 for the detailed description.**

## 5 Results

### 5.1 Groundwater model calibration and validation

Steady-state hydraulic heads were calibrated against historical GWL observations from 47 monitoring points in Brandenburg (LfU, 2023). The calibrated parameters included hydraulic conductivity and storativity, with the best-estimate values provided in Table A1. RMSE after calibration was 6.6 m, which we find acceptable for a regional-scale model of this complexity (Fig. 5a).

Available temperature data are sparser (14 locations) with a large gap between shallow groundwater monitoring points and deep petroleum exploration boreholes. RMSE for temperature for the steady-state case was 3.0 °C (Fig. 5b).

Transient hydraulic heads were validated against historical GWL time series (Fig. 6). The model successfully reproduced the magnitude of the historical GWL decline between 1980 and 2021. Proper validation of transient subsurface temperatures was not feasible due to the lack of repeated long-term measurements in areas not affected by urban heat island effects and uncertainties in sensor depth relative to the water table.



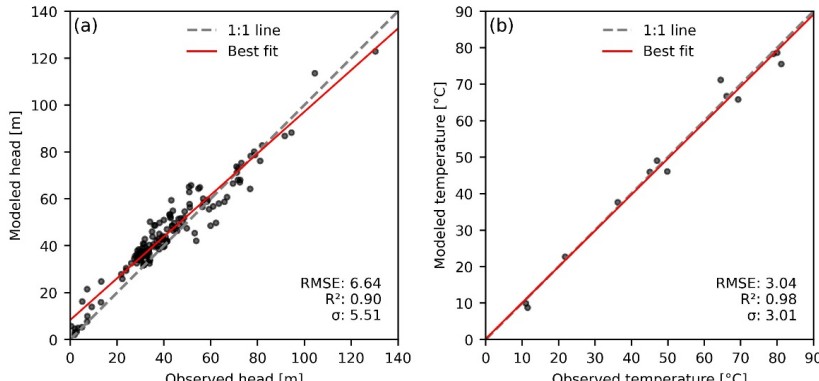

**Figure 5: Steady-state groundwater model calibration results: simulated versus observed hydraulic head and temperature at monitoring points (locations shown in Figure 1c. $R^2$ – coefficient of determination; $\sigma$ – standard deviation.**

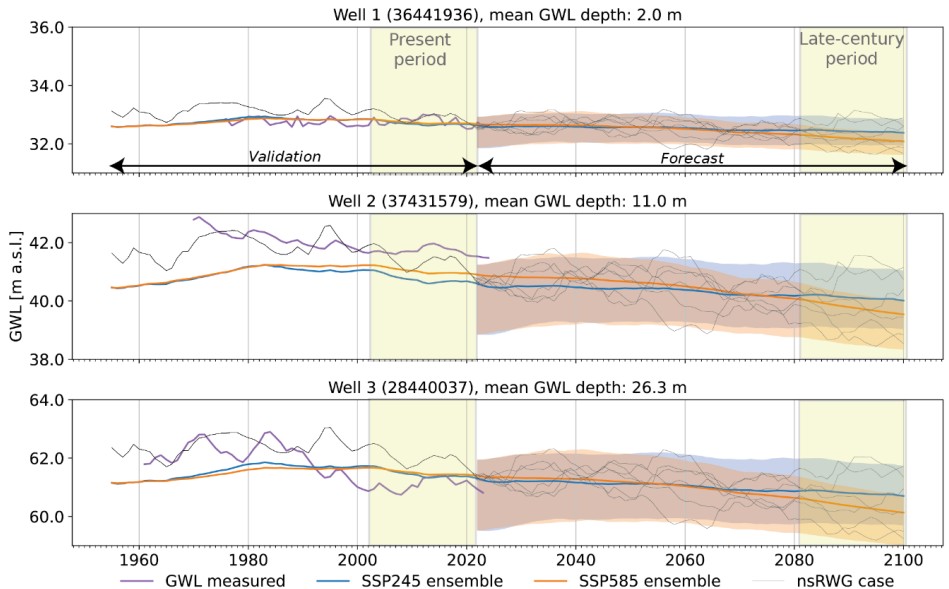

**Figure 6: GWL time series for three monitoring wells (locations shown in Fig. 7a). Colored lines indicate ensemble mean, shaded areas represent P10-P90 range of the ensemble distribution, black lines are examples of individual nsRWG realizations. Numbers in brackets denote well ID from the state groundwater monitoring network (LfU, 2023).**

## 5.2 Hydraulic heads

The projected evolution of hydraulic heads between the present and late-century periods closely follows trends imposed by changes in groundwater recharge. In both climate scenarios, the largest long-term changes are observed at watershed divides, whereas the lowest changes occur adjacent to river valleys (Fig. 7a). Under SSP585, head declines at watershed divides reach up to 4.8 m, compared to 3.6 m under SSP245. Minimal changes in lowlands can be linked to artificially-regulated water levels in rivers, making groundwater less responsive to
recharge variability.



For the SSP585 ensemble mean, the heads are projected to decline at approximately the same rate as observed during the historical period, while under SSP245, the rate of decline slows down after 2070 (Fig. 6). Simulated heads for discrete nsRWG cases reveal a potential temporary reversal in head trends, with head increases lasting up to 10 years (Fig. 6). Such behavior has been frequently observed in the historical data and attributed to teleconnections with inter-annual climatic oscillations (Liesch and Wunsch, 2019).

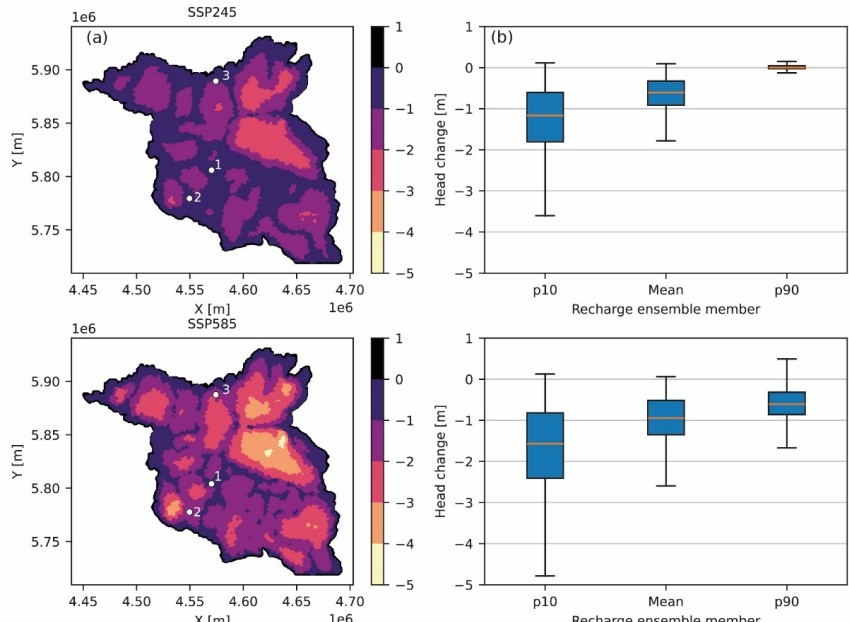

**Figure 7: Projected changes in hydraulic heads between the present and late-century periods: (a) spatial distribution of head changes for the SSP ensemble mean; coordinates are shown in DHDN / 3-degree Gauss-Kruger zone 4 (EPSG:31468); labels refer to wells, shown in Fig. 6; (b) comparison of P10, mean, and P90 ranges.**

### 5.3 Subsurface thermal field

Vertical temperature profiles at four representative locations, each characterized by different local geology and hydraulic conditions, reveal spatial variability in the geothermal gradient, depth of the seasonal envelope, and the extent of thermal disturbance from the projected global warming (Fig. 8).

The first-order difference in modeled geotherms, reaching 11 °C at 1000 m between the selected locations, is driven by the basal heat flux, which varies due to heterogeneities in the underlying geological units, such as depth of the Moho, sediment thickness, and the distribution of salt structures (Noack et al., 2012).

Profiles A, B, and D exhibit relatively low geothermal gradients in the upper 100–200 m, suggesting a strong influence of recharge and downward advection of cold meteoric water. A distinct inflection at 150–200 m coincides with the depth of the Rupelian aquitard. In profile D, penetration of advective cooling and the change in gradient slope are deeper due to local erosion of the Rupelian (Fig. 8a). Profile A, located on a glacial plateau, has a shallow seasonal envelope, entirely above the water table, suggesting no advective transfer of the seasonal thermal signal into the saturated zone. Profile C, extracted from a river valley, shows a higher geothermal gradient




and a smaller magnitude of seasonal fluctuations, reflecting discharge conditions with some flow contribution
from the deeper aquifers.

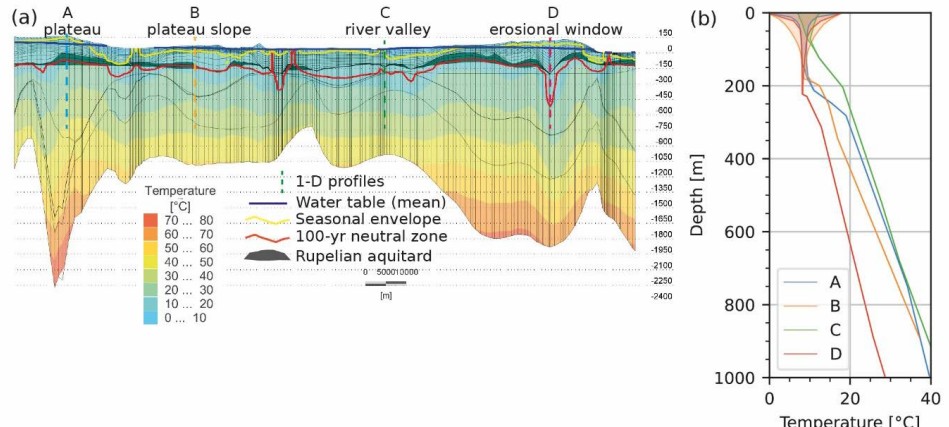

**Figure 8: (a) cross-section of the initial thermal field, showing variability in the maximum depth of thermal disturbance (100-yr neutral zone) and a seasonal envelope; (b) initial temperature profiles (lines) and seasonal envelopes (shaded areas). The location of the cross-section is shown in Fig. 1d.**

To isolate the contribution from the changing climate, we examined the difference in temperature and groundwater flux between the present and late-century periods (Fig. 9).

The groundwater flux in the Quaternary unit ranges from $10^{-8}$ to $10^{-6}$ m s$^{-1}$. Projected changes in the vertical component of the groundwater flux ($q_z$) in the late-century period relative to the present period are $\pm 2 \times 10^{-9}$ m s$^{-1}$ or 5–40 %. (Fig. 9a). In the saturated part of the Quaternary, decreasing recharge (SSP245/585) reduces downward flux at locations A, B, and D and correspondingly lowers upward flux for location C (and vice versa for increasing recharge scenarios, M1/M2).

Climate-induced thermal disturbance penetrates the first few hundred meters, being controlled by the depth of the Rupelian aquitard in locations A and B (Fig. 9b). $Pe$ number in the Quaternary unit ranges between $10^{1}$–$10^{2}$, pointing to advective heat transport in the interval of meteoric water circulation, whereas in the pre-Rupelian section the range is $10^{-4}$–$10^{-1}$, suggesting a conduction-dominated regime (Domenico and Schwartz, 1997). In location C, temperature disturbance does not reach the Rupelian due to the thicker post-Rupelian section. In location D, where the Rupelian is locally eroded, the warming signal penetrates into the pre-Rupelian strata.

Projected groundwater warming at the water table reaches 2.5 °C and 1.1 °C under SSP585 and SSP245, respectively, except at location C, where surface warming is hampered by upward flow. In contrast, the isolated effect of recharge is weaker, as seen in profiles A and D, where higher recharge scenario (M2), despite a larger fraction of winter recharge, results a temperature difference relative to SSP585 not exceeding 0.4 °C.



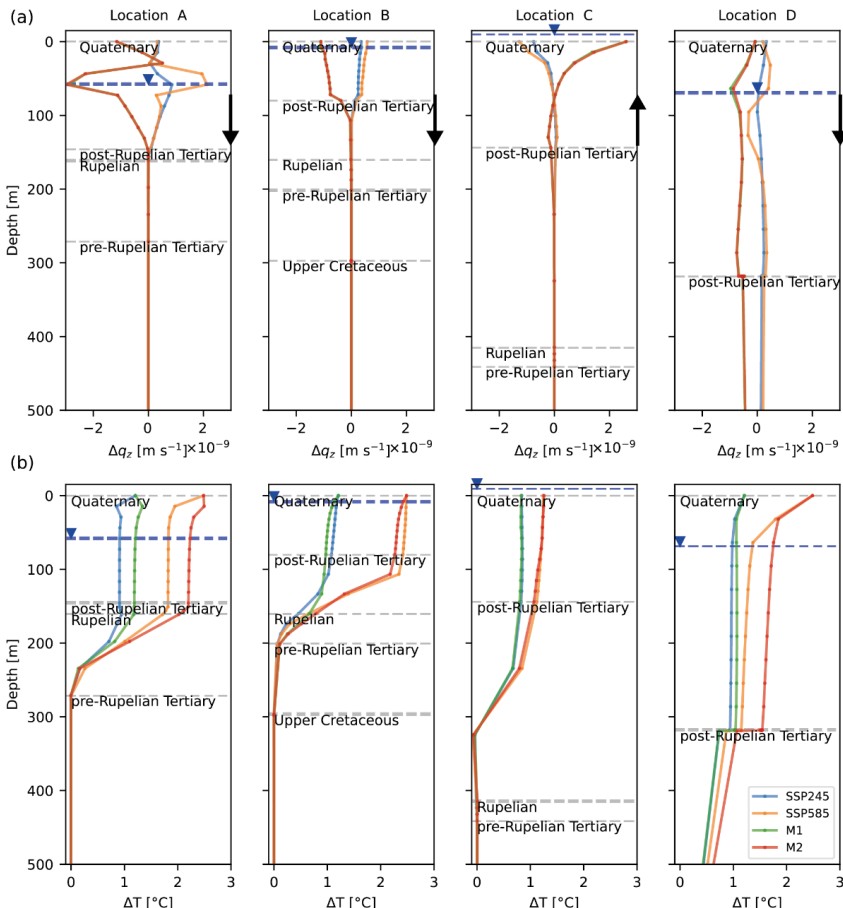

**Figure 9: Changes in (a) vertical groundwater flux and (b) temperature vs depth between the present and late-century periods. The scenario pairs SSP245/M1 and SSP585/ M2 assume the same degrees of surface warming but decreasing/increasing groundwater recharge, respectively. The blue line indicates water table depth. The arrow shows the direction of vertical groundwater flow ($q_z$ vector is always oriented upward).**

We derive maps of the stable long-term subsurface temperatures by extracting the depth below which the temperature difference between the present and late-century periods is <0.2 °C (Fig. 10a). This steady temperature level varies spatially, being shallowest (<200m) along river valleys and in areas of thin sedimentary cover along the southern model boundary. The deepest penetration of the warming signal (450–550 m) occurs in the northwestern part of the area within the Quaternary tunnel valleys and in the east, where the Rupelian aquitard is largely absent (Fig. 1d). Under SSP585 the depth of the surface warming propagation is on average 50 m deeper than under SSP245 (Fig. 10b). Changes in recharge flux alone, in contrast, have a negligible impact on this depth. The depth of the seasonal envelope is expected to rise under decreased downward flux. Combined with projected water table decline under SSP585, this would expand over time the modeled area where seasonal temperature fluctuations are fully damped before reaching the water table from 5 % to 9 % (Fig. A4).





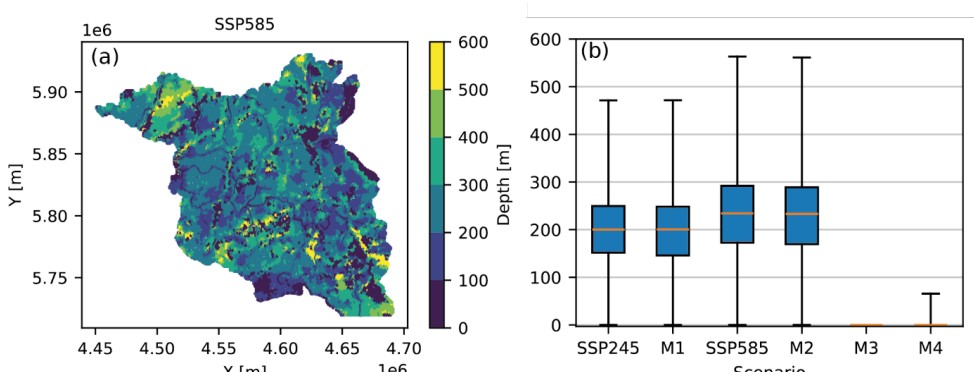

**Figure 10: Depth to long-term steady temperatures: (a) spatial variability under SSP585; (b) comparison of the depth variability between scenarios. Scenarios M3 and M4 assume changes in recharge and constant surface temperatures.**

### 5.4 Accumulated heat

Figure 11 illustrates the spatial distribution of the accumulated subsurface heat due to climate change and compares ranges between the simulated scenarios. Surface temperature rise exerts a first-order control on the amount of incremental accumulated thermal energy. Under SSP585, mean accumulated heat is more than twice that of SSP245 (736 vs 327 MJ m⁻²). The groundwater recharge effect is significantly weaker: a change from declining recharge to increasing annual/winter recharge led to almost no difference in mean accumulated heat for

SSP245 warming levels, and resulted in a 5 % increase for SSP585, and a slight reduction when steady temperatures are assumed.

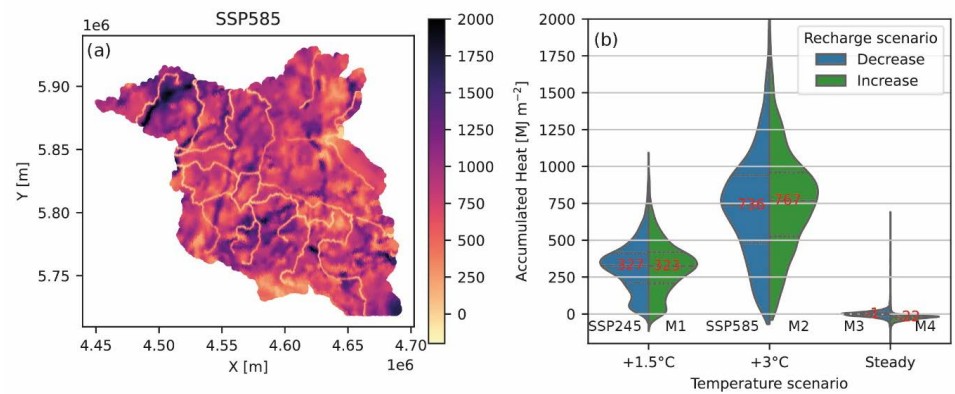

**Figure 11: Accumulated heat-in-place: (a) map for scenario SSP585, (b) violin plots (equal-width-normalized) comparing probability distributions of accumulated heat-in-place per m² between the scenarios. Red labels indicate**
**P50 values; black labels refer to model scenarios.**

The majority heat is being accumulated in the Quaternary unit, with the contribution from other stratigraphic units decreasing with depth (Fig. 12). Within individual units, the spatial distribution of accumulated heat is strongly controlled by the gross thickness variability. In the Quaternary, the largest gains are localized in thick SW-NE-
450 oriented tunnel valleys. In contrast, the subsurface beneath river valleys does not accumulate much heat, because



upward flow and groundwater discharge supress surface heat transfer. In the post-Rupelian Tertiary unit, the most heat in place is accumulated in the northwest of the study area, consistent with the basin-wide stratigraphic thickening trend. In the pre-Rupelian Tertiary and Upper Cretaceous units, climate-driven heat accumulation is scattered, being tied to areas where the overlying Rupelian aquitard has been eroded, and where the thickness is increased, associated with syn-tectonic sedimentation in rim synclines of Zechstein salt diapirs.

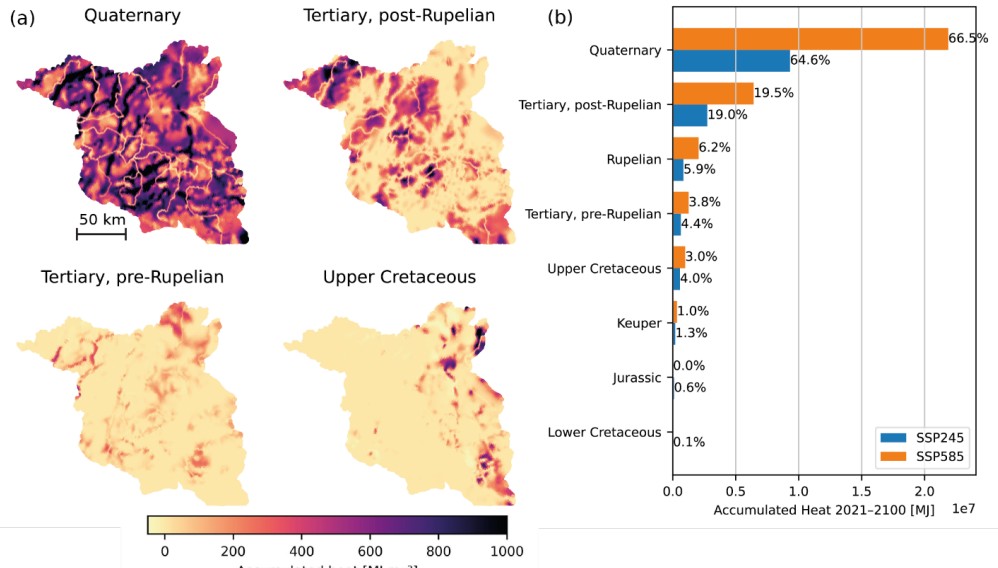

**Figure 12: Accumulated heat by geological unit between the present and late-century periods: (a) maps for scenario SSP585, (b) comparison between SSP585 and SSP245 with percent labels indicating unit's contribution to the total accumulated heat.**

## 6 Discussion

### 6.1 Intrinsic controls on groundwater dynamics and thermal field

Spatial analysis of the groundwater modeling results presented in Sect. 5 allows us to distinguish several surface and geological domains that have a distinct control on flow and heat transport in the post-Rupelian section and, accordingly, portray different feedback effects to climate change (Fig. 13). We briefly characterize them below for further referencing in the subsequent discussion.

**Surface domains (SD):**

- **SD1 – glacial plateaus,** characterized by a deep unsaturated zone (depth to water table > 10 m). Annual fluctuations in surface temperature and precipitation are effectively damped before reaching the phreatic zone. Instead, the aquifers respond primarily to inter-annual and long-term climatic perturbations. After reaching the water table, the climate signal may be transmitted by advection laterally but also vertically down to >200 m. Notably, these fluxes are driven not by higher recharge rates, but by steep topographic gradients: SD1 is characterized by the highest magnitudes of advective overprinting of the conductive thermal field.



-    **SD2 – plateau slopes** with a shallow water table (< 10 m). A thin unsaturated zone allows both annual and inter-annual recharge and temperature fluctuations to reach the unconfined aquifer. Groundwater flow has a strong lateral component, directing the thermal signal from recharge areas toward discharge areas along the flow paths.

     -    **SD3 – floodplains and river valleys** with a water table at or near the surface. Groundwater flow has a
strong upward component. Surface temperature fluctuations, both annual and long-term, are modulated by discharging groundwater. Discharge temperatures depend on the contribution of deeper groundwater, originating at depths, where temperature remains stable over time (if geology allows for such connection). Upward advection leads to relatively higher geothermal gradients.

**Geological domains (GD):**

-    **GD1 – less than 100 m-thick Quaternary, underlain by the Rupelian.** In such configuration, the Rupelian Clay acts as both a hydraulic and thermal barrier, isolating pre-Rupelian aquifers from climatic perturbations within the simulated timeframe. Groundwater dynamics is characterized by shallow circulation, with a predominantly lateral flow component towards streams, and a greater sensitivity to recharge variability due to a lower storage.

-    **GD2 – 100–500 m-thick Quaternary, underlain by the Rupelian.** As such, permeable Quaternary sediments essentially accumulate all the excess heat from surface warming. The temperature disturbance does not reach the Rupelian aquitard within the modeled period.

     -    **GD3 – 100–500 m-thick Quaternary, incising into the Rupelian.** In this configuration, the Rupelian is locally eroded, allowing direct hydraulic connection between the Quaternary aquifers and the deeper
formations. The crossflow direction depends on the spatial relationship with surface domains. If GD3 is found below SD1, surface warming propagates deep into the Tertiary units due to downward advective flux. When GD3 underlies SD3, conditions favor the influence from deep groundwater, e.g., preferential warm water discharge.

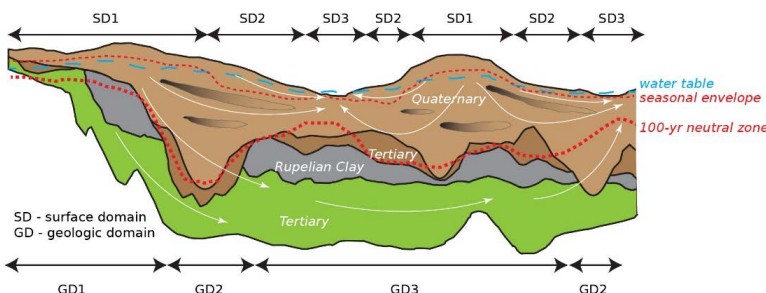


**Figure 13: Schematic cross-section, illustrating patterns of groundwater flow and the thermal field for different surface and geological domains (see text for the domains' description).**

### 6.2     Effects of recharge projections on groundwater dynamics

Changes in groundwater recharge directly affect the storage of unconfined layers. Projected recharge decreases under SSP245 and SSP585 ensembles mean cause hydraulic head declines of 0.1-1.8 m and 0.5-2.6 m across



Brandenburg, respectively (Fig. 7b). To a lesser extent, this results in a reduction of groundwater flux due to a preferential decline of heads on glacial plateaus (SD1) and consequently a lower hydraulic gradient. Beyond this hydrodynamic effect, it remains uncertain which surface domains experience greater reduction in net recharge:

SD2 and SD3, where groundwater is depleted due to evaporation from the water table and root transpiration (Francke and Heistermann, 2025) or SD1, where an already thick unsaturated zone further accumulates multi-year drought effects (Tsypin et al., 2025).

In addition to water table elevation, changes in recharge rates have been shown to affect modeled groundwater travel times, flow rates, and stream discharge (Jing et al., 2020; Dams et al., 2012), although there has been no

evidence of modifications to the regional flow regime (i.e., groundwater trajectories and catchment boundaries). Our results also indicate that the tested magnitude of projected recharge changes (-20 to +50 %) only influences basin-scale flow to a limited extent, that is, the groundwater system is resilient and reversible. Initial hydraulic gradients ranged between $4 \times 10^{-4}$ and $2 \times 10^{-3}$, while projected heads under SSP585 reduce them only by 5-10 %. More fundamental changes in groundwater dynamics due to climatic variability would require a sustained forcing

extended beyond the IPCC's 2100 horizon. In the absence of long-term climatic projections, modeling such changes over thousands of years to millennia typically relies on synthetic long-term scenarios, often inferred from paleoclimatic trends. Climatic cycles with wet and dry periods may result in amplitudes of GWL fluctuations up to 80 m (Zhou et al., 2020). Emplacement and subsequent retreat of continental ice sheets forced large pressure redistribution leading to reorganization or even a reversal of basin-scale groundwater flow (Frick et al., 2022a;

Lemieux et al., 2008; Boulton et al., 1996).

The sustained decline in heads on plateaus relative to floodplains and river valleys, if exceeds 10–20 m, would approach magnitudes of the local terrain rise (for Brandenburg), being able to modify the basin-wide groundwater dynamics. A reduction of hydraulic gradients leads to a smoother potentiometric surface, which has been shown to suppress local groundwater circulation and enhance regional flow systems (Havril et al., 2018). This is

manifested by longer flow paths between areas of recharge and discharge, as the potentiometric surface falls below local drainage levels. To investigate such changes, attention must be given to the assumed density of the river network and river stage levels, since the inclusion of only the highest-order streams tends to enhance the regional flow systems (Haitjema and Mitchell-Bruker, 2005) and influence the resulting advection circulation patterns.

### 6.3    Effects of recharge projections on the thermal field

As shown in Sect. 5.3, sustained surface warming under the presence of groundwater advection is projected to affect subsurface temperatures down to depths of 100–550 m by the end of the century. While the mean depth of penetration depends on the magnitude of implied surface warming, lateral variations in advection rates can be explained by hydraulic gradients, varied between surface domains and hydraulic conductivity distribution, controlled by geological domains.


**Heat advection**

The different sensitivity of surface domains to climate change supports the conclusions of Burns et al. (2017) and Taniguchi (2021) that this response is amplified in topographically elevated recharge areas (SD1), and subdued in low-lying discharge areas (SD3). This spatial differentiation is driven by the downward and upward components

of the regional groundwater flow field, respectively.



Under the same degree of surface warming, it is the difference in recharge projections that impacts advection in Cenozoic units (SSP245 vs M1, SSP585 vs M2, M3 vs M4). However, scenarios with higher total recharge and a higher fraction of winter recharge (M1, M2, M4) did not reveal a reversal or a widespread buffering of the groundwater warming.

Present-day seasonality, with most recharge occurring between December and May, results in a mean annual recharge temperature approximately 3 °C lower than the mean surface temperature. However, when the effect of global warming is excluded, the isolated effect of further shifting recharge from warm to cold months leads to only a 0.1-0.2 °C reduction (Fig. A3), since current recharge is already winter-dominated (Fig. 3c).

A higher total recharge transfers more heat to the subsurface in the recharge zones compared with declining
recharge scenarios, which becomes more evident under the stronger warming levels, leading to up to 0.4 °C of additional increase locally. Under steady-state conditions, advection-driven "cooling" may reach tens of degrees (Noack et al., 2013). However, in a transient model, subsurface cooling would require an increased advective flux to reach conduction-dominated reservoirs, transporting colder than in situ fluid, despite the parallel effects of surface warming. This is, however, unlikely because subsurface temperatures are more sensitive to changes in
surface temperature than to changes in recharge, as inferred from 1-D analytical solutions of advection-diffusion sensitivity to upper boundary conditions (Kurylyk et al., 2017).

Thus, while advection plays a major role in heat transport in the first hundred meters, local enhancements and buffering of the climatic signal spatially compensate each other, irrespective of the recharge trend. This implies that projected changes in advection rates and seasonality have only minor effects on the basin-wide heat budget
and can counteract the regional net heat gain from the surface warming only to a limited extent (Fig. 11b).

These findings contrast with the results by Epting et al. (2021), who, in their modeling study of Swiss alluvial aquifers, inferred that under a high-end climate change scenario the cooling effect of increased winter recharge may overcome the warming due to rising surface temperatures. One possible explanation lies in the different hydrogeological conditions between the two study areas: in the study by Epting et al. (2021) groundwater is largely
fed from rivers, potentially leading to a more direct response to runoff seasonality. This highlights how such feedback depends on local conditions, such as the type of infiltration (precipitation vs snowmelt and/or effluent conditions), vadose zone thickness and the presence of preferential flow pathways, and the water table type (recharge-controlled vs topography-controlled).

**Hydraulic conductivity distribution**

The downward component of groundwater flow is sensitive to the hydrostratification, particularly to the presence of units impermeable to flow. These aquitards force a shift in heat transport mechanism from advection to conduction, as inferred from $Pe$ values, and additionally enhance lateral groundwater flow. In our regional model, the penetration depth of the warming signal largely coincides with the first resolved hydraulic barrier, the Rupelian
Clay, at depths of 50–500 m, provided it is sufficiently thick. Potential exceptions occur due to the cross-flow through erosional windows in the Rupelian aquitard, allowing the global warming signal to reach underlying aquifers within the simulated timeframe (Fig. 12a).

Local-scale models that additionally consider second-order heterogeneities within the Quaternary unit have demonstrated how intermediate aquitards or transitions between reservoir facies further control shallow-to-deep
groundwater interactions (manifested, for example, in saltwater upconing) and lateral flow patterns (Chabab et



al., 2022; Stackebrandt and Franke, 2015; Baldwin and Sprenger, 2024). This highlights that the degree of heterogeneity resolved by a model, particularly the depth and continuity of the first hydraulic barrier, is an important factor in the simulated extent of the subsurface response to climatic forcing.

**Water table elevation**

Given that the amplitude of the annual surface temperature cycle remains steady, the depth of the annual thermal envelope responds to changes in groundwater flux in a similar way to the penetration of the long-term warming discussed above. The regional decline of the water table combined with a reduced vertical flux is expected to increase temperature damping at the water table, almost doubling the extent of the unconfined aquifer not

subjected to annual variability under SSP585 (Fig. A4). Moreover, a water table decline below a certain level may deactivate the feedback mechanism between groundwater and the land-surface energy fluxes by restraining evapotranspiration from the saturated zone (Maxwell and Kollet, 2008).

Since the impact of evolving surface conditions was simulated in the groundwater model at monthly time steps, this study did not address the influence of the frequency and intensity of extreme precipitation events, which are

projected to increase by up to 30 % in the study region (Nguyen et al., 2024; Pfeifer et al., 2021). Whether a reduced diffuse recharge from percolation via the unsaturated zone may enhance the role of heat conduction, yet intermittent extreme precipitation events could still generate rapid preferential flow. Simulating the impact of preferential flow on the heat transport would require adding new physics or discrete elements to the simulation and much shorter, hourly to daily time steps (Beven and Germann, 2013).

**6.4     Implications for geothermal energy utilization**

We have demonstrated that SSP245 and SSP585 scenarios lead to a mean net increase of 330 and 740 MJ m$^{-2}$ of stored thermal energy in the aquifers between the present and late-century periods, respectively (Fig. 11). The estimated accumulated heat in the upper 100 m amounts to 180–380 MJ m$^{-2}$, which is comparable to the 130–250 MJ m$^{-2}$ projected by Benz et al. (2024) for the same region as part of their global assessment. Further refinement

of heat density distribution could be achieved by considering facies heterogeneity and accounting only for net reservoir volume within stratigraphic units (Frick et al., 2022b).

The accumulated thermal energy has the potential to be recycled in shallow geothermal systems, such as GSHP or borehole thermal energy storage (BTES). In the target installation interval between the water table and the depth of 100 m, our model predicts a mean increase of 1.5 °C for SSP585 (0.8 °C for SSP245). The projected

accumulated heat in this interval is worth several months to a year of a single household's space heating consumption in Germany (Odyssee-Mure, 2024).

In urban environments, waste heat may contribute 40–50 % more to the accumulated subsurface heat compared to natural settings (Menberg et al., 2013; Rivera et al., 2017; Bayer et al., 2016). Surface temperature in the Berlin metropolitan area is up to 3 °C higher than in predominantly rural Brandenburg, leading to an incremental heat

accumulation below the city (SenStadt, 2020; Tsypin et al., 2024). The assigned surface temperature BC for our study was assumed uniform in space. A potential effect of the additional variability associated with the urban heat island effect and a surface sealing is shown in Appendix B.

Despite projected aquifer net heat gains, climate change and urbanization can degrade the performance of open systems, such as LT-ATES: increased cooling demand in summer would result in a disproportional expansion of



the warm plume, leading to a strong decline in cold well performance, driven by a mismatch between injection
       and production rates (Godinaud et al., 2025). Aquifers with high ambient groundwater velocities ($>3\times10^{-4}$ m s$^{-1}$)
       pose a risk for additional thermal energy losses in ATES systems due to groundwater displacement (Bloemendal
       and Hartog, 2018), while in closed systems, such as GSHP, elevated groundwater fluxes above $1\times10^{-7}$ m s$^{-1}$
       typically enhance subsurface heat exchange capacity, helping to sustain operating temperatures (Abesser et al.,
630    2023).

       Considering a typical 30-year lifespan of shallow geothermal installations, modeled change in ambient
       groundwater flux may reach ±10 %, depending on the recharge scenario. This points to the relevance of including
       at least surface temperature BC and ideally transient recharge or hydraulic head BC when modeling a system's
       performance during the design phase to account for the additional impact of climate evolution under specific local
settings.

## 7    Conclusions

       We investigated the joint effects of surface warming and evolving recharge patterns on regional groundwater
       dynamics and subsurface heat transport for an area of 25,000 km$^2$ in Central Europe. The advantage of testing
       these effects with regional groundwater models is the ability to simulate complex 3-D flow patterns emerging
from the interplay of topographic gradients and a heterogeneous permeability field.

       Active shallow groundwater circulation in Quaternary fluvioglacial deposits is responsible for a deeper
       propagation of the surface warming signal on glacial plateaus, while buffering climatic impacts in floodplains and
       river valleys. The first regional aquitard sets the ultimate depth limit of advective overprint, except where it is
       eroded.

The magnitude of groundwater temperature increase depends primarily on the global warming scenario: the mean
       sensitivity to surface temperature is approximately 1.4 °C, while the difference between recharge scenarios
       contributes up to 0.4 °C between the present (2002-2021) and late-century (2081-2100) periods. A projected late-
       century decrease in annual recharge of 10–20 % or even a hypothetical increase in the share of colder winter
       recharge has no capacity to reverse the ongoing groundwater warming trend. A decrease of vertical groundwater
flux may locally buffer the progradation of the surface warming, however this will have minimal net effect on the
       total accumulated subsurface heat due to the long-term counterbalance of advective effects in recharge versus
       discharge zones. Related to this is a projected decline in hydraulic heads of up to 2–5 m under SSP585, leading
       to greater damping of the seasonal temperature fluctuations at the water table.

       While global-scale assessments focus on conductive surface-groundwater heat exchange, local differences in
hydraulic gradients can significantly modulate spatial patterns of subsurface warming. Repeated temperature logs
       are rarely able to test variations between recharge and discharge areas, or they are acquired in urban areas where
       recharge effects are hindered. Extending monitoring networks to link groundwater warming with landscape
       hydrology can help to advance field evidence of the coupled effects discussed in this study.

       Our study suggests that while future recharge rates will undoubtedly have direct implications on groundwater
availability, they should also be taken into account when assessing the shallow geothermal potential and
       temperature-dependent ecosystem under climate change.



**Appendix A: Additional figures and tables**

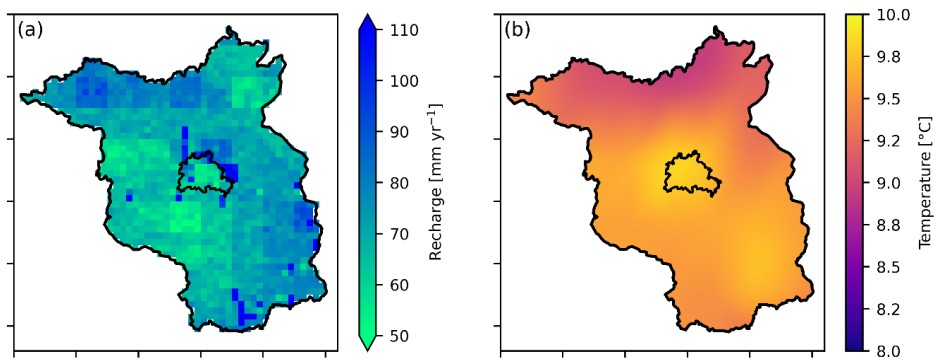

**Figure A1: Variability of groundwater recharge and near-surface air temperature across the study area for the present period.**




**Figure A2: Histograms of mean annual recharge for Brandenburg, showing the distribution of 50 nsRWG realizations between the present and late-century periods for each GCM. Left: SSP245; right: SSP585.**

10000




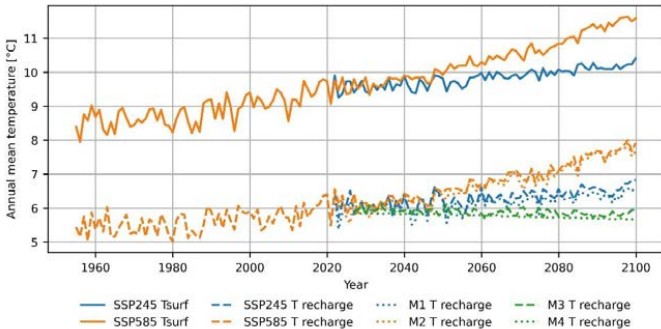

**Figure A3: Comparison between mean surface temperature and recharge temperatures, with the latter derived as a weighted average of monthly surface temperatures by groundwater recharge rate.**

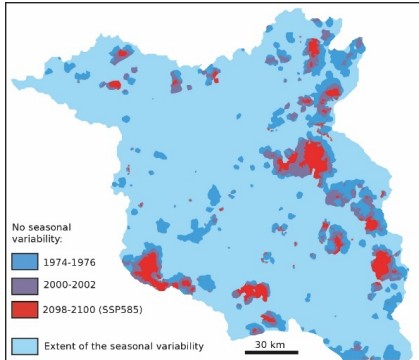

**Figure A4: Extent of seasonal temperature damping at the water table during three periods. As a threshold for damping, we used a magnitude of temperature fluctuations <1 °C.**


**Table A1: Hydraulic and thermal properties of the groundwater model units: K – hydraulic conductivity, φ – porosity, $S_y$ – specific yield, $S_s$ – specific storage, $C$ – volumetric heat capacity, $\lambda$ – thermal conductivity, and $Q_r$ – heat production.**

| Unit | $K_{xy}$ [m s$^{-1}$] | $K_z$ [m s$^{-1}$] | Φ | $S_y$ / $S_s$ [m$^{-1}$] | $C$ [J m$^{-3}$ K$^{-1}$] | $\lambda$ [W m$^{-1}$ K$^{-1}$] | $Q_r$ (W m$^{-3}$) |
|---|---|---|---|---|---|---|---|
| Quaternary | $2.1 \times 10^{-4}$ | $1.2 \times 10^{-7}$ | 0.23 | $0.1 / 1 \times 10^{-4}$ | $1.7 \times 10^{-6}$ | 1.5 | $7.0 \times 10^{-7}$ |
| post-Rupelian Tertiary | $5.8 \times 10^{-7}$ | $5.8 \times 10^{-8}$ | 0.23 | $/ 1 \times 10^{-4}$ | $1.7 \times 10^{-6}$ | 1.5 | $7.0 \times 10^{-7}$ |
| Rupelian | $1.2 \times 10^{-9}$ | $1.2 \times 10^{-10}$ | 0.2 | $/ 1 \times 10^{-4}$ | $1.8 \times 10^{-6}$ | 1 | $4.5 \times 10^{-7}$ |
| pre-Rupelian Tertiary | $1.2 \times 10^{-4}$ | $1.2 \times 10^{-8}$ | 0.1 | $/ 1 \times 10^{-4}$ | $1.7 \times 10^{-6}$ | 1.9 | $3.0 \times 10^{-7}$ |
| Upper Cretaceous | $1.2 \times 10^{-6}$ | $1.2 \times 10^{-7}$ | 0.1 | $/ 1 \times 10^{-4}$ | $2.3 \times 10^{-6}$ | 1.9 | $3.0 \times 10^{-7}$ |
| Lower Cretaceous | $1.2 \times 10^{-6}$ | $1.2 \times 10^{-7}$ | 0.13 | $/ 1 \times 10^{-4}$ | $2.3 \times 10^{-6}$ | 2 | $1.4 \times 10^{-6}$ |
| Jurassic | $1.2 \times 10^{-6}$ | $1.2 \times 10^{-7}$ | 0.13 | $/ 1 \times 10^{-4}$ | $2.2 \times 10^{-6}$ | 2 | $1.4 \times 10^{-6}$ |
| Keuper | $1.2 \times 10^{-7}$ | $1.2 \times 10^{-8}$ | 0.06 | $/ 1 \times 10^{-4}$ | $2.3 \times 10^{-6}$ | 2.3 | $1.4 \times 10^{-6}$ |
| Water | | | | | $4.2 \times 10^{-6}$ | 0.65 | 0 |



### Appendix B. Subsurface warming in urban conditions

Urban environments exhibit distinct peculiarities in their shallow subsurface PT field due to excess heat from the urban climate and infrastructure, extensive surface sealing, altered soil structure, and controlled water levels. Although the study area is predominantly rural, with agricultural and natural land cover, approximately 4 % is occupied by the city of Berlin with the artificial land cover.

Most of the repeated vertical temperature profiles suitable for model validation were obtained within the city limits. The measurements obtained over the past four decades showcase a reversal of the geothermal gradient, associated with the anthropogenic warming (Fig. B1). In the study area, an inflection point, where the temperature gradient shifts from negative to positive, commonly occurs at 60–70 m depth. The maximum depth of the recorded temperature increase occurs at 90–120 m (SenStadt, 2020). The shape of the temperature profiles indicates gradual, conduction-dominated groundwater warming with low advection rates due to low percolation rates and enhanced surface heating.

We conducted a sensitivity realization in the groundwater model by locally adjusting BCs and calibrating model properties within the urban domain. The following modifications were assumed for the model inputs:

– Surface temperature forcing adjusted to reach a mean annual temperature of +11.1 °C by 2024, compared to +9.7 °C for the entire study area.

– Groundwater recharge reduced by a factor of 10 to represent surface sealing and.

– Quaternary permeability in selected layers reduced by a factor of 100 to represent intermediate aquitards.

This model scenario was simulated for the period 1955-2024. Modeled 1-D temperature profiles demonstrate overall good consistency with the measured data in depths of the inflection point and penetration of long-term thermal disturbance (Fig. B1). Lower absolute temperatures relative to Distributed Temperature Sensing (DTS) data can be due to local heat sources and DTS interpretation uncertainty.

Overall, this sensitivity scenario confirms higher historical rates of warming within the subsurface urban heat island under conditions of impeded recharge and additional heat inputs. Such a trend presents both an opportunity for waste heat recovery and a risk to groundwater quality and replenishment. Further improvement in reproducing the absolute magnitude of subsurface warming would require development of localized models with refined geological structure, parametrization, and a higher mesh resolution.



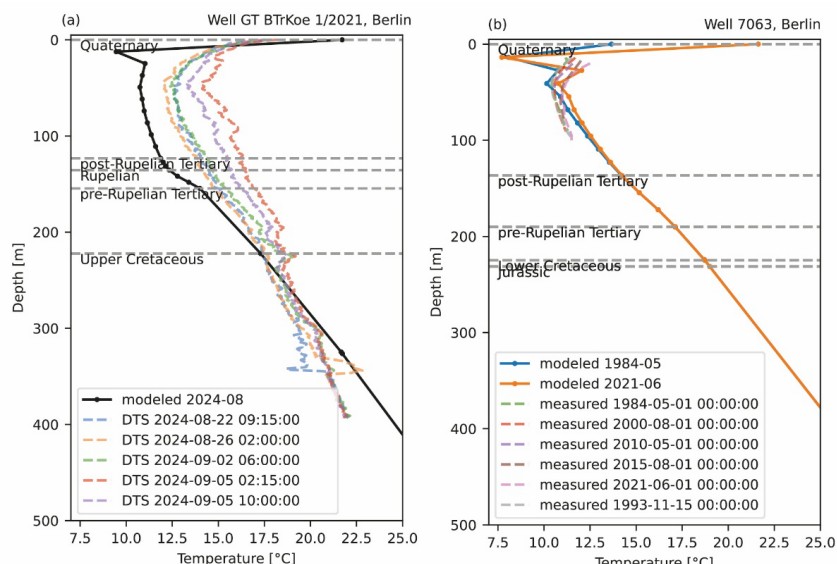

**Figure B1: Comparison between the simulated temperature profile (solid line) and (a) distributed temperature sensing**
**(DTS) fiber-optic cable measurements in the ATES research well in Adlershof, Berlin and (b) temperature logging in**
**a monitoring well in Neukölln, Berlin (dashed lines). Temperature difference in repeated DTS profiles result from a**
**cementation job and other technical interferences. Marked geological units are derived from the geological model.**
**Measured data is taken from Pei et al. (2024), Singer (2025), and SenStadt (2020).**

### Data availability

The simulated nsRWG weather forcings, mHM recharge, and groundwater heads and temperatures are available for download via associated data publication (https://dataservices.gfz-potsdam.de/panmetaworks/review/f623d9bc294e3125862c0b0946043c09f5463eb3210b171324c5dc88139cd0f4/ or https://doi.org/10.5880/GFZ.CUEG.2025.001).

### Author Contribution

MT, VDN, and MC conceptualized the study and developed the methodology, with further contributions from GB and EL. VDN conducted weather generator and hydrologic model simulations. MT carried out groundwater modeling, performed the formal analysis, and prepared the visualization. MSW and CK provided supervision. MT prepared the original draft of the manuscript, with all co-authors contributing to the review and editing.

### Competing interests

The authors declare that they have no conflict of interest.



**Acknowledgements**

This work utilized high-performance computing resources made possible by funding from the Ministry of Science, Research and Culture of the State of Brandenburg (MWFK). An academic license for FEFLOW software was kindly provided by DHI A/G. The style and grammar of the manuscript were improved using AI tools (OpenAI, Perplexity AI).

The authors would like to thank Andreas Güntner and Mark Grmek for their valuable discussions.

**Financial Support**

MT received financial support for the research and publication of this article from GFZ Helmholtz Centre for Geosciences. VDN was supported by the German Federal Ministry of Education and Research (projects 01LP1903E and 01LP2324E) within the ClimXtreme network (FONA3). Open Access funding enabled and organized by Project DEAL.

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
