# Peer review of "Influence of groundwater recharge projections on climatedriven subsurface warming: insights from numerical modeling"

_EGUsphere, 2025_

## Author Comment (AC1)

Tsypin and colleagues have submitted a manuscript on evolving subsurface thermal regimes, with the key novel contribution being the role of changing recharge. They conclude that advection plays a role in general, but that changing advection (e.g. more cold-season recharge) cannot overwhelm the overall pattern of groundwater warming from diffusion. The study is generally well written and interesting.

We would like to thank reviewer#1 for their positive feedback on our study, as well as for the in-depth review and constructive suggestions. In what follows, we provide our point-by-point responses to each comment (our answers are marked in red).

A few comments, in no particular order:

1. Many times in this manuscript the authors use 'temperature' to refer (I think) to surface air temperature; such examples can often be found when referring to climate data. But in a manuscript focused on groundwater temperature, they should always be clear about where in the earth system domain they are referring to when they mention temperature.

We thank reviewer#1 for this comment. In the corrected version of the manuscript, we will specify the type of temperature being referred to.

Reviewer#1 correctly points out that in some instances, especially in the context of the weather generator and recharge modeling, we refer to near-surface air temperature, measured at 1-2 m above the ground, as simply "temperature". This indeed may be confusing, as the manuscript primarily focuses on subsurface temperature. In our study, we used the same near-surface air temperature both for calculating potential evapotranspiration and as the upper boundary condition for advective-conductive heat transport.

Regarding top boundary conditions, we have come across various approaches: assigning surface air temperature directly (Bense and Kurylyk, 2017), using surface air temperature with an offset correction (Bense et al., 2017), using shallow soil temperature (Benz et al., 2024), or using land surface temperature from remote sensing (Casillas-Trasvina et al., 2022). We elaborate on our choice for using near-surface air temperature later in this reply (Question 7c).

2. The authors miss some quite related studies from the Netherlands on the role of groundwater flux on subsurface warming. For example Bense et al. (10.1029/2019WR026913) show how *changes to groundwater flow regimes* can be inferred from temperature profiles (a close topic to the present manuscript although this former study did not forecast into the future), and Bense et al. 2017 (10.1002/2017GL076015) showed that recharge influences where the inflection point occurs in evolving temperature profiles (see L542, where I think a conference abstract is cited for this point). Also, Figure 9 of the present study shows deep temperature shifts when the flow regime changes. This was also demonstrated by Bense et al. 2017 (10.1002/2017WR021496) who showed that using a linear temperature-depth profile as an initial condition results in deep shifts in temperature with time because that initial condition doesn't have the same groundwater flow regime as the forward modeling (e.g., Taniguchi et al. 1999 solution). Side note: I think the authors could do a better job of explaining that groundwater flow regime shifts are what is driving

the temperature changes at depth in some of their results (which could not happen under heat diffusion alone under the timescales considered here).

We appreciate the suggestions of the related studies. We were surprised to realize that we originally didn't cite any works of Victor Bense, despite being familiar with them.

Bense et al. (2020) is a valuable study, as it includes a rare dataset of repeated deep temperature profiles and demonstrates temperature changes due to shifts in groundwater dynamics rather than due to heat diffusion, though the described groundwater flux changes were caused by groundwater abstraction, not due to climate change (see also recent example by Klepikova et al. (2025)). A higher impact of multi-year pumping than of ongoing climate-driven recharge reduction on groundwater dynamics was also shown for the study region by Tsypin et al. (2024).

Monitoring the depth of the inflection point proved to be a powerful tool for quantifying groundwater flow rates in Bense and Kurylyk (2017). Unfortunately, the vertical resolution of our regional 3D groundwater model does not allow to simulate changes in the inflection point depth with time.

For the calculation of the vertical groundwater flux rates from temperature profiles we cited the classic works of Stallman (1965) and Bredehoeft and Papadopulos (1965). Bense et al. (2017) presents a major advancement, focusing on both transient effects of temperature and groundwater regime. In our study, we account for this by including recharge and surface air temperature history since 1950 to simulate current and future groundwater dynamics.

Regarding the side note, we think that Figure 9 in the manuscript effectively conveys the relationship between flow regime (expressed as downward groundwater flux) and temperature changes. We agree that heat diffusion alone cannot drive these changes, but the opposite is also true: changes in only recharge/advection rates without surface temperature shifts are also not sufficient to explain magnitude of temperature changes in the subsurface.

We will include some of these references in the revised manuscript.

3. Figure 1b - it is stated that this is a 3D view of the model mesh. However, neither the mesh nor the model parameters are indicated (from what I can see). This is just a geological model (or perhaps geometric model of the model domain). Much is made of the geology in this paper (rightfully so), but I think it would be useful to just indicate the parameters on this model domain (at least k and Ss - or put in a clear table). If a table, the thermal properties for each layer would be useful to list too (in the main text).

Figure 1b was a screenshot of the FEM mesh, colored with different material regions (i.e., stratigraphic units). The low image resolution in the preprint and the small size of mesh elements makes mesh geometry barely visible. Only the mesh refinement along rivers can be recognized. We will make sure that it will have higher resolution in the corrected manuscript (Figure 1).

Instead of indicating individual parameters on the model domain (this would be not practical, given their number), we chose to color the model by stratigraphic layers and provide Table A1 with the thermal and hydraulic properties of each layer, enabling a straightforward mapping. Additionally, the model can be downloaded in the linked data publication.

Figure 1: 3-D view of the FEM mesh.

Table 1: Hydraulic and thermal properties of the groundwater model units: K – hydraulic conductivity,  $\phi$  – porosity,  $S_y$  – specific yield,  $S_s$  – specific storage, C – volumetric heat capacity,  $\lambda$  – thermal conductivity, and  $Q_r$  – heat production. Properties were kept identical across scenarios to isolate climatic controls and were taken from previous studies (Scheck and Bayer, 1999; Noack et al., 2013) except Quaternary and Tertiary post-Rupelian units, which were subject to calibration. Note that the original manuscript wrongly had a minus sign shown for thermal conductivity values.

| Unit                   | K xy [m s -1 ] | K z [m s -1 ] | ф    | S y / S s [m -1 ] | C [J m -3 K -1 ] | λ [W m -1 K -1 ] | Q r (W m -3 ) |
|------------------------|--------------------------------------|-------------------------------------|------|----------------------------------------------------|----------------------------------------|----------------------------------------|-------------------------------------|
| Quaternary             | 2.1 × 10 -4               | 1.2 × 10 -7              | 0.23 | 0.1 / 1 × 10 -4                         | 1.7 × 10 6                  | 1.5                                    | 7.0 × 10 -7              |
| post-Rupelian Tertiary | $5.8 \times 10^{-7}$                 | $5.8 \times 10^{-8}$                | 0.23 | $/ 1 \times 10^{-4}$                               | $1.7 \times 10^{6}$                    | 1.5                                    | $7.0 \times 10^{-7}$                |
| Rupelian               | 1.2 × 10 -9               | 1.2 × 10 -10             | 0.2  | $/ 1 \times 10^{-4}$                               | $1.8 \times 10^{6}$                    | 1                                      | $4.5 \times 10^{-7}$                |
| pre-Rupelian Tertiary  | $1.2 \times 10^{-4}$                 | 1.2 × 10 -8              | 0.1  | $/ 1 \times 10^{-4}$                               | $1.7 \times 10^{6}$                    | 1.9                                    | $3.0 \times 10^{-7}$                |
| Upper Cretaceous       | $1.2 \times 10^{-6}$                 | $1.2 \times 10^{-7}$                | 0.1  | $/ 1 \times 10^{-4}$                               | $2.3 \times 10^{6}$                    | 1.9                                    | $3.0 \times 10^{-7}$                |
| Lower Cretaceous       | $1.2 \times 10^{-6}$                 | $1.2 \times 10^{-7}$                | 0.13 | $/ 1 \times 10^{-4}$                               | $2.3 \times 10^{6}$                    | 2                                      | $1.4 \times 10^{-6}$                |
| Jurassic               | $1.2 \times 10^{-6}$                 | $1.2 \times 10^{-7}$                | 0.13 | $/ 1 \times 10^{-4}$                               | $2.2 \times 10^{6}$                    | 2                                      | $1.4 \times 10^{-6}$                |
| Keuper                 | $1.2 \times 10^{-7}$                 | 1.2 × 10 -8              | 0.06 | $/ 1 \times 10^{-4}$                               | $2.3 \times 10^{6}$                    | 2.3                                    | $1.4 \times 10^{-6}$                |
|                        |                                      |                                     |      |                                                    |                                        |                                        |                                     |
| Water                  |                                      |                                     |      |                                                    | 4.2 × 10 6                  | 0.65                                   | 0                                   |

For reviewer's #1 reference we show the 3D distribution of some properties below (Figure 2).

Figure 2: Distribution of hydraulic conductivity, thermal conductivity of solid, and volumetric heat capacity of solid in the model.

4. The recharge modeling is impressive - it might be good to be more explicit that all of these recharge realizations do not translate into equivalent model runs for groundwater temperature. I only figured that out later, but I admit to reading this in a moving car with children yelling in my ear etc. The authors calibrate this to streamflow. How does the model performance compare for just baseflow? That might be more comforting to know, since that is a better indicator of whether the recharge and discharge processes are being captured reasonably.

Reviewer#1 correctly points out that for groundwater simulations, we used not every 750 recharge realizations, but a mean, p90, p10 ensemble members for each of the two emission scenarios.

Most of the hydrological model calibration was presented in an earlier German-wide study by Guse et al. (2024). The model performs well for high-flow periods, as its original focus was flood analysis. The highest Nash–Sutcliffe model efficiency coefficient was obtained for gauges on major rivers such as Elbe and Rhein, indicating that regional trends are well captured.

We have analyzed the calculated baseflow, and in the corrected version of the manuscript we will comment on the model performance for this variable within the study region.

5. Eq. 1 - units don't work - is Recharge in units of 1/s (like a volume flux per unit volume)?

**Yes R should be $[s^{-1}]$ . We thank reviewer#1 for the correction.**

6. This is a big numerical model domain with many elements. Was this solved with high-performance computing or anything - or just run on a 'regular' computer? My simulations with this many nodes took weeks to run, but that was 10 years ago.

This question was also raised by reviewer#2. Below we provide detailed information about the simulations and will include a concise description in the updated manuscript.

Computational experiments for the climate and hydrological components—including the non-stationary weather generator, the mHM hydrological model, and the workflow connecting both—were performed on the GLIC high-performance computing (HPC) system at the GFZ German Research Centre for Geosciences. The system uses the SLURM workload manager, a standard HPC scheduling environment that assigns resources and coordinates parallel workloads. This setup allowed us to run the modelling chain efficiently across 50 compute nodes in parallel. Each node provided 40 GB of RAM, and each task used four CPU cores. With this configuration, the complete workflow—from the nsRWG simulations to the mHM impact modelling—required roughly 10 days of wall-clock time.

The groundwater model was simulated separately on a local workstation equipped with an Intel Core Ultra 7 155H processor (16 cores) and 32 GB of RAM. Each of the six scenarios required ~3,500 computational timesteps to cover a 145-year period with monthly boundary conditions, resulting in simulation times between 13 and 26 hours.

7. Boundary conditions questions: a) is there any vertical hydraulic gradient on the vertical boundary conditions at the sides? b) Does the model allow for exfiltration at the surface if the water table rises too high? c) I'm surprised the surface air temp and ground surface temperature are so similar. Is there no snow here? The mean annual air temp of 9C suggests there would be.

We would like to clarify our choice of boundary conditions:

- a) No, the hydraulic head was constant with depth along the sides of the model. Therefore, a vertical pressure gradient according to the gravitational effect is present, which does not generate an additional vertical flow component. This may imply that groundwater flow / discharge into rivers at the boundaries is underestimated. While in the presented model the side edges were kept open, in a previous work we demonstrated the effect of no-flow boundaries for the same model domain (Tsypin et al., 2024). An additional complication is that flow in the deep aquifers is controlled more by the structural dip of the sedimentary layers than by local drainage configuration. In follow-up studies, a more granular boundary condition assignment could be tested, combining no-flow, river, and in/out flux boundary conditions.
- b) In the "phreatic" model definition of the unconfined aquifer, the water level may theoretically rise above the ground surface. This would have two consequences. (1) Along all major river valleys, a Cauchy boundary condition was prescribed, allowing

- for groundwater exfiltration. (2) In other areas, the initial water table was normally several meters deep. Therefore, even in periods of highest recharge, the amount of unaccounted outflow due to discharge into ephemeral streams and/or direct evapotranspiration from the water table is expected to be low.
- c) Yes, the small difference between surface temperature and near-surface air temperature also surprised us (left image below). However, this difference refers to modeled temperatures from the GCM within the IPCC6 report. We have looked more carefully at historical data since then (right image). The mean difference is still relatively small, 1.2°C, which can be explained by following:
  - a. According to the German Weather Service, the annual mean number of snowing days in 1985-2015 was 21, and the number of days with snow cover >1 cm was 24. Only January has long-term mean temperature slightly below zero.
  - b. The study uses monthly temperatures, while the highest difference between air and surface temperatures occurs on the diurnal cycle
  - c. The analysis was performed for natural land cover. In urban environments, using air temperatures likely underestimates the degree of surface warming, as discussed in Appendix B of the original manuscript.

Figure 3: relationship between simulated surface air temperature (TAS) and surface temperature (TS) in Brandenburg during 2015-2100 period from the climate model Amon\_MPI-ESM1-2-HR\_ssp585\_r1i1p1f1\_gn (in Kelvin) (left); historical data from Potsdam secular station (1994-2024) (https://opendata.dwd.de/) (right).

8. Using mesh size for characteristic length for Peclet number is arbitrary (but maybe common) - how do the authors know what the threshold is for this Pe formulation for which advection matters? Later on they present Pe values and use these to highlight the role of advection, but this is not a formulation for which Pe =1 implies that advection = conduction (like say the Bredehoeft and Papadopulos 1965 formulation does - if I remember right). So what is the threshold for when advection matters? Maybe that is hard to say.

According to the classic approach, where characteristic length is defined by travel distance, Pe values greater than 1 indicate a dominance of the advection, while Pe values smaller than 1 indicate diffusion-dominated transport (Bredehoeft and Papadopulos (1965), DOMENICO

and PALCIAUSKAS (1973) for heat transport or Ogata and Banks (1961) for mass transport). Using mesh element size as the characteristic length for Pe is common in numerical simulators (FEFLOW, GOLEM, OpenGeoSys) with a primary relevance for assessing numerical stability of advection-dispersion: spatial discretization in advection dominated systems often leads to artificial oscillations and numerical dispersion (Huysmans and Dassargues, 2005).

Caution is required when interpreting the role of advection, because element size may vary with mesh discretization and between layers of different thickness. Therefore, we should limit interpretation of Pe to a qualitative level. For example, in a S-N model cross-section in Figure 4 below, a relatively higher role of advection is characteristic of shallower, more permeable aquifers, as well as for glacial plateaus with higher hydraulic gradients.

After considering the reviewer's comment, we decided to exclude the paragraph on Pe interpretation in the revised manuscript.

Figure 4: Model S-N cross-section with Pe numbers.

9. The recharge change runs in the groundwater model are interesting (see Table 1), but I'm still not clear where some of these came from (esp. the discrete synthetic scenario). I guess at least one is more of a sensitivity run rather than an expected scenario? Also, it would be helpful if Table 1 were more precise about timelines rather than "present" and "late century". What years are the before/after runs here for?

The information on the exact time ranges and derivation of the synthetic scenario is given earlier (L245-249) and later (Section 4.2) in the text, respectively. We agree with reviewer#1 that it is needed to be marked in the table as well.

The corrected table is given in Table 2 below.

Table 2: Summary of the tested groundwater modeling scenarios.

| Scenario                                                | Description                                 | ΔT (surface)            | Δ Groundwater recharge              | Comment                                         |  |  |  |  |
|---------------------------------------------------------|---------------------------------------------|-------------------------|-------------------------------------|-------------------------------------------------|--|--|--|--|
| Present (2002-2021) to late-century (2081-2100) periods |                                             |                         |                                     |                                                 |  |  |  |  |
|                                                         | Ensemble sc                                 | enarios derived with n  | sRWG and mHM (Section 4.1)          |                                                 |  |  |  |  |
| SSP245                                                  | Moderate emission pathway                   | +1.5 °C -10 %           |                                     | Ensemble mean                                   |  |  |  |  |
| SSP585 High-end emission pathway                        |                                             | +3 °C                   | -20 %                               |                                                 |  |  |  |  |
|                                                         | Syn                                         | thetic and reference sc | enarios (Section 4.2)               |                                                 |  |  |  |  |
| M1                                                      | Recharge increase,                          | +1.5 °C                 | +50 %, higher winter recharge ratio |                                                 |  |  |  |  |
| moderate warming                                        |                                             |                         | , 6                                 | after Marx et al.                               |  |  |  |  |
| M2                                                      | Recharge increase, strong warming           | +3 °C                   | +50 %, higher winter recharge ratio | (2021)                                          |  |  |  |  |
| M3                                                      | Recharge as SSP585, present-day temperature | 0 °C                    | -20 %                               | Reference cases to isolate the recharge effects |  |  |  |  |
| M4                                                      | Recharge increase, present-day temperature  | 0 °C                    | +50 % Winter                        |                                                 |  |  |  |  |

10. Model calibration - consider presenting the RMSE normalized to head range. I think that would do a better job of showing the fit. 6.6 m sounds high, but the head is highly variable, so it is not a bad fit at all.

Figure 5 below shows the updated plots with added RMSE normalized to the range, as well as bias value.

Figure 5: Steady-state groundwater model calibration results: simulated versus observed hydraulic head and temperature at monitoring points (locations shown in Error! Reference source not found.c). RMSE – Root Mean Square Error, nRMSE – normalized to value range;  $R^2$  – coefficient of determination;  $\sigma$  – standard deviation.

11. Where do groundwater temp measurements come from for calibration? Case borehole temp profiles? Pumped groundwater with temp recorded? Open boreholes? Large diameter? Are convection or seasonal bias concerns?

Temperatures from deep aquifers (>500 m) that we used to validate the initial thermal field, are a mix of (a) bottom-hole temperatures (BHT) obtained in wells shortly after drilling, thus representing perturbed borehole temperatures and (b) continuous temperature logs, mostly measured after the boreholes were cased (Förster, 2001). All BHT temperatures were previously corrected by Horner-plot correction and the exponential integral method based on a model simulating the temperature build-up during shut-in time of a well (Förster, 2001). The estimated error of the corrected measurements was  $\pm 3^{\circ}$ C or  $\pm 10^{\circ}$ C. Some temperature logs were also corrected using a simple empirical correction, with the error in final estimates of  $\pm 3^{\circ}$ C (Förster, 2001).

Shallow groundwater temperature monitoring is available from a handful of wells in Berlin and is limited to 80 m below ground (SenStadt, 2020). Unfortunately, continuous vertical profiles and Distributed Temperature Sensing (DTS) are only available for the urban area. We chose to present them separately (Appendix B), due to strong land cover overprint.

12. Figure 7 - it would be interesting to also see changes in discharge presented somehow. The authors talk about changes in recharge (input) and head (storage), but what about output (relevant for some of the reasons groundwater temperature is relevant)

Following the reviewer's suggestion, we will comment on discharge changes in the revised manuscript. A more in-depth analysis would require more thorough parametrization of rivers (colmation layer conductivity, streambed geometry), and is better suited to a catchment scale study (e.g., Haacke et al. (2018)). We can recommend a recent paper by Li et al. (2025), who used similar modeling tools to infer climate-driven river network contraction.

13. L386 "Profile A, located on a glacial plateau, has a shallow seasonal envelope, entirely above the water table, suggesting no advective transfer of the seasonal thermal signal into the saturated zone" - I don't think you have to be saturated for heat advection to matter. It depends on the water flux, not the saturation per se. if you have recharge through the vadose zone, advection can still theoretically matter.

We completely agree with this comment. Our language is surely confusing here. We did not want to say that there is no heat advection through the vadose zone. The point is that the vadose zone on plateaus can be so thick that even with the vertical water flux (relatively slow), the seasonal temperature fluctuations are fully attenuated above the water table. This results in stable year-round temperatures of the phreatic aquifer, though still subject to a long-term warming.

Saturation levels are relevant here because the numerical code scales down hydraulic conductivity proportionally to saturation above the water table.

Minor comments

L60-65 - I think the controversy over the role of advection is overstated here. In the Benz et al. study, the authors used a parsimonious approach in a global study and justified this using 2D numerical models of advection and conduction. They basically found that for typical hydrogeological systems and recharge rates, heat advection is not a primary driver of groundwater temperature change. However, they note that is not the case for some basins. Other studies cited here are in steeper topography or with much higher recharge, and so it makes sense that they found that advection mattered.

We thank reviewer#1 for this comment. It would indeed be more precise to call it a spectrum of findings rather than a controversy. However, we also think the contrasting conclusions on the role of heat advection came not only due to basin-specific geology, recharge and topography, but also an applied approach (e.g., inclusion of lateral groundwater flow, unsaturated zone assumptions, and boundary conditions).

Figure 11 a - label missing for scale bar

The color bar and the vertical axis of Figure 11 b have the same range and share the same label. The same is true for Figures 10 and 7. In order to avoid any misinterpretation we are going to mention this in captions of the corrected manuscript.

Conclusions \_ 'depth limit of advection overprint' sounds a bit jargony for the conclusions

Agreed. Proposed change: The first regional aquitard imposes the lower limit on downward advective heat transport, except where it is eroded.

In general, these comments are all easy to address I think, and I look forward to seeing this article published.

**References:**

Bense, V. F. and Kurylyk, B. L.: Tracking the Subsurface Signal of Decadal Climate Warming to Quantify Vertical Groundwater Flow Rates, Geophysical Research Letters, 44, 12,244-212,253, <a href="https://doi.org/10.1002/2017GL076015">https://doi.org/10.1002/2017GL076015</a>, 2017.

Bense, V. F., Kurylyk, B. L., de Bruin, J. G. H., and Visser, P.: Repeated Subsurface Thermal Profiling to Reveal Temporal Variability in Deep Groundwater Flow Conditions, Water Resources Research, 56, e2019WR026913, <a href="https://doi.org/10.1029/2019WR026913">https://doi.org/10.1029/2019WR026913</a>, 2020.

Bense, V. F., Kurylyk, B. L., van Daal, J., van der Ploeg, M. J., and Carey, S. K.: Interpreting Repeated Temperature-Depth Profiles for Groundwater Flow, Water Resources Research, 53, 8639-8647, https://doi.org/10.1002/2017WR021496, 2017.

Benz, S. A., Irvine, D. J., Rau, G. C., Bayer, P., Menberg, K., Blum, P., Jamieson, R. C., Griebler, C., and Kurylyk, B. L.: Global groundwater warming due to climate change, Nature Geoscience, 17, 545-551, <a href="https://doi.org/10.1038/s41561-024-01453-x">https://doi.org/10.1038/s41561-024-01453-x</a>, 2024.

Bredehoeft, J. D. and Papadopulos, I. S.: Rates of vertical groundwater movement estimated from the Earth's thermal profile, Water Resources Research, 1, 325-328, <a href="https://doi.org/10.1029/WR001i002p00325">https://doi.org/10.1029/WR001i002p00325</a>, 1965.

Casillas-Trasvina, A., Rogiers, B., Beerten, K., Wouters, L., and Walraevens, K.: Characterizing groundwater heat transport in a complex lowland aquifer using paleo-temperature reconstruction, satellite data, temperature—depth profiles, and numerical models, Hydrol. Earth Syst. Sci., 26, 5577-5604, <a href="https://doi.org/10.5194/hess-26-5577-2022">https://doi.org/10.5194/hess-26-5577-2022</a>, 2022.

DOMENICO, P. A. and PALCIAUSKAS, V. V.: Theoretical Analysis of Forced Convective Heat Transfer in Regional Ground-Water Flow, GSA Bulletin, 84, 3803-3814, 10.1130/0016-7606(1973)84<3803:Taofch>2.0.Co;2, 1973.

Förster, A.: Analysis of borehole temperature data in the Northeast German Basin: continuous logs versus bottom-hole temperatures, Petroleum Geoscience, 7, 241-254, 10.1144/petgeo.7.3.241, 2001.

Guse, B., Han, L., Kumar, R., Rakovec, O., Luedtke, S., Herzog, A., Thober, S., Samaniego, L., and Wagener, T.: Spatio-Temporal Consistency and Variability in Parameter Dominance on Simulated Hydrological Fluxes and State Variables, Water Resources Research, 60, e2023WR036822, https://doi.org/10.1029/2023WR036822, 2024.

Haacke, N., Frick, M., Scheck-Wenderoth, M., Schneider, M., and Cacace, M.: 3-D Simulations of Groundwater Utilization in an Urban Catchment of Berlin, Germany, Adv. Geosci., 45, 177-184, 10.5194/adgeo-45-177-2018, 2018.

Huysmans, M. and Dassargues, A.: Review of the use of Péclet numbers to determine the relative importance of advection and diffusion in low permeability environments, Hydrogeology Journal, 13, 895-904, 10.1007/s10040-004-0387-4, 2005.

Klepikova, M., Bense, V., Le Borgne, T., Guihéneuf, N., and Bour, O.: Impact of groundwater extraction on subsurface thermal regimes, Environmental Research Letters, 20, 054048, 10.1088/1748-9326/adc8bb, 2025.

Li, Y., Yang, X., Lischeid, G., Wollheim, W. M., Jomaa, S., Zhou, X., and Rode, M.: Responses of Wetted River Network Contraction and Expansion Dynamics to Prolonged Drought, Water Resources Research, 61, e2024WR038938, <a href="https://doi.org/10.1029/2024WR038938">https://doi.org/10.1029/2024WR038938</a>, 2025.

Marx, A., Boeing, F., Rakovec, O., Müller, S., Can, Ö., Malla, C., Peichl, M., and Samaniego, L.: Auswirkungen des Klimawandels auf Wasserbedarf und -dargebot, WASSERWIRTSCHAFT, 111, 14-19, 10.1007/s35147-021-0905-5, 2021.

Noack, V., Scheck-Wenderoth, M., Cacace, M., and Schneider, M.: Influence of fluid flow on the regional thermal field: results from 3D numerical modelling for the area of Brandenburg (North German Basin), Environmental earth sciences, 70, 3523-3544, <a href="https://doi.org/10.1007/s12665-013-2438-4">https://doi.org/10.1007/s12665-013-2438-4</a>, 2013.

Ogata, A. and Banks, R. B.: A solution of the differential equation of longitudinal dispersion in porous media: fluid movement in earth materials, US Government Printing Office, 1961.

Scheck, M. and Bayer, U.: Evolution of the Northeast German Basin — inferences from a 3D structural model and subsidence analysis, Tectonophysics, 313, 145-169, <a href="https://doi.org/10.1016/S0040-1951(99)00194-8">https://doi.org/10.1016/S0040-1951(99)00194-8</a>, 1999.

SenStadt: Senatsverwaltung für Stadt-entwicklung, Bauen und Wohnen - Berlin Environmental Atlas, Berlin, <a href="https://www.berlin.de/umweltatlas/en/water/groundwater-temperature/">https://www.berlin.de/umweltatlas/en/water/groundwater-temperature/</a>, 2020.

Stallman, R. W.: Steady one-dimensional fluid flow in a semi-infinite porous medium with sinusoidal surface temperature, Journal of Geophysical Research (1896-1977), 70, 2821-2827, <a href="https://doi.org/10.1029/JZ070i012p02821">https://doi.org/10.1029/JZ070i012p02821</a>, 1965.

Tsypin, M., Cacace, M., Guse, B., Güntner, A., and Scheck-Wenderoth, M.: Modeling the influence of climate on groundwater flow and heat regime in Brandenburg (Germany), Frontiers in Water, 6, <a href="https://doi.org/10.3389/frwa.2024.1353394">https://doi.org/10.3389/frwa.2024.1353394</a>, 2024.

---

## Author Comment (AC2)

I enjoyed reading the manuscript "Influence of groundwater recharge projections on climate-driven subsurface warming: insights from numerical modeling." The manuscript presents a clear objective, employs a sound methodology to achieve it, and reports results that are well presented and supported by the discussion and conclusions. Most of my comments focus on improving clarity in certain sections.

We would like to thank reviewer#2 for their positive feedback on our study, as well as for the in-depth review and constructive suggestions. In what follows, we provide our point-by-point responses to each comment (our answers are marked in red).

Line 167: Could you please explain why these seven GCMs were selected? Since many GCMs are available, I am curious whether these models have specific characteristics that make them particularly suitable for this study. Clarifying this would also address the question of why seven—why not five, ten, or the full ensemble?

Thank you for this helpful comment. The selection of the seven GCMs in our study follows the same strategy used in Nguyen et al. (2024), where models were chosen based on a combination of performance-and independence-based criteria. In that work, GCMs were evaluated using the ClimWIP (Climate model Weighting by Independence and Performance) method (Brunner et al., 2020) as implemented in ESMValTool v2.6.0 (Eyring et al., 2016). The selection relied on quantitative metrics—specifically each model's distance to ERA5 over Europe for 1985–2014 temperature and sea-level pressure climatology, annual variability, and temperature trends—as well as qualitative considerations of model independence and spread following the recommendations of Merrifield et al. (2023). This approach identifies a subset of CMIP6 models that provides a representative range of climate responses while avoiding unnecessary interdependencies between closely related models and reducing the computational load for both the weather generator and subsequent impact modeling.

We will clarify this briefly in the updated manuscript.

Figure 1: Performance weights for 15 GCMs resulting from the ClimWIP procedure based on the preselected evaluation criteria for the historical period (modified after Nguyen et al., 2024). 7 GCMs selected for the current study are marked in red.

For completeness, could you report the area of the model domain? The mesh spans  $170 \times 150$  km, but (as I understand it) the numerical model only simulates part of that rectangle. Are some cells within the bounding rectangle not active? A brief clarification would help.

The FEM mesh was not rectangular. The provided dimensions are average extents in X and Y direction. The actual shape of the FEM model is irregular, as shown in Figures 1b and 1c. The model boundary follows topographic elements (major rivers and divides), largely corresponding to the boundaries of the Brandenburg federal land. None of the cells were deactivated. We will specify the area of the model domain in the corrected manuscript (27.6 thousand km2).

It would be helpful to include a short paragraph or a few sentences describing the computational cost of the simulations and the computing resources used (e.g., HPC system, number of cores, total runtime).

This suggestion was also given by reviewer#1. Below is the detailed information about the simulations. We will include a concise description in the updated manuscript.

Computational experiments for the climate and hydrological components—including the non-stationary weather generator, the mHM hydrological model, and the workflow connecting both—were performed on

the GLIC high-performance computing (HPC) system at the GFZ German Research Centre for Geosciences. The system uses the SLURM workload manager, a standard HPC scheduling environment that assigns resources and coordinates parallel workloads. This setup allowed us to run the modelling chain efficiently across 50 compute nodes in parallel. Each node provided 40 GB of RAM, and each task used four CPU cores. With this configuration, the complete workflow—from the nsRWG simulations to the mHM impact modelling—required roughly 10 days of wall-clock time.

The groundwater model was simulated separately on a local workstation equipped with an Intel Core Ultra 7 155H processor (16 cores) and 32 GB of RAM. Each of the six scenarios required ~3,500 computational timesteps to cover a 145-year period with monthly boundary conditions, resulting in simulation times between 13 and 26 hours.

Line 219: Why were thermal and hydraulic parameters calibrated for only two layers? A short justification would improve clarity.

We agree that the revised manuscript would benefit from such justification. In general, there are two reasons:

- 1. The upper two units (Quaternary and post-Rupelian Tertiary) are the most relevant for the study objective, since they directly experience the impact of the varying upper boundary conditions. The influence on the deeper aquifers is local and depends mostly on the thickness distribution of the Rupelian aquitard, which is known fairly well and doesn't require addition calibration. Moreover, above the Rupelian aquitard advection dominates the heat transport, whereas below, heat conduction plays the main role. Therefore, the calibration was performed for the parameters of the advective diffusive domain, which matter more for the time scale of the model.
- 2. The necessary information about the deeper units is sparser. Selected constant effective rock properties for the deeper units are consistent with published parametrized models of the same area (Noack et al., 2013; Frick et al., 2019). In these papers, sensitivity studies on some key parameters have been already tackled, e.g., effective permeability of stratigraphic units.

**Line 227: What proportion (in percentage) of the model's lateral boundaries coincide with rivers?**

The proportion of the river boundaries is approximately 50% (basically eastern and northwestern edges corresponding to Oder and Elbe rivers respectively). In the revised manuscript we will replace the phrase "At lateral model edges a constant-head Type I (Dirichlet) boundary condition (BC) was assigned, given that they largely correspond to major rivers" with are more quantitative statement.

Figure 2: River network of the study area (LfU, 2023) and the model boundaries.

Some darker points appear in Figure 5a. Do these points represent a specific subset of observations, or are they a rendering artifact? Additionally, could you add a measure of bias between modeled and observed heads and temperatures?

Yes, this is an artefact of the overlapping points. The bias for heads was 3.70, and for temperature it was -0.42. We thank reviewer#2 for these suggestions. The new version of the plot reflects these changes (Figure 3).

Figure 3: Steady-state groundwater model calibration results: simulated versus observed hydraulic head and temperature at monitoring points (locations shown in Figure 1c). RMSE – Root Mean Square Error, nRMSE – normalized to value range; R2 – coefficient of determination;  $\sigma$  – standard deviation.

In Figure 6 why were these three wells chosen? Would it not be more informative to select wells from three distinct regions with different expected drawdown responses (Fig. 7a)?

These three wells were selected to illustrate typical pathways of GWL evolution for wells with different initial water table depths: < 5 m, 5-20 m, and >20 m. The proposal of reviewer#2 is also valid. However, we have two limitations: (1) the regions in Figure 7 do not overlap between scenarios; (2) there are no suitable wells from the regions with the highest drawdown range with adequately long coverage of the historic record. Therefore, we propose to keep the wells presented in the original manuscript.

Line 386: The text states that Profile A has a shallow seasonal envelope entirely above the water table, implying no advective transfer of the seasonal signal into the saturated zone. Isn't this also the case for Profile D?

Yes, it is also the case for the Profile D, although the main reason for including Profile D was to demonstrate the thermo-hydraulic effect of eroded Rupelian in the text and later in Figure 9. The text will be adjusted accordingly.

Section 6.1 is clearly written, and Figure 13 is well designed. However, I'm a bit confused: GD3 is described as an area where "the Rupelian is locally eroded, allowing direct hydraulic connection between Quaternary

aquifers and deeper formations". In Figure 13, the region labeled GD3 appears to still contain Rupelian clay (gray unit), which does not reflect such a direct connection. Could you clarify this?

We thank reviewer#2 for pointing on this inconsistency. GD2 and GD3 labels must be swapped in the schematic (Figure 4) to be consistent with the description in the text.

Figure 4: Schematic cross-section, illustrating patterns of groundwater flow and the thermal field for different surface and geological domains.

Line 517: The sentence "The groundwater system is ... reversible" may cause confusion. It is clear that your model does not simulate mechanical deformation of the subsurface (and this is okay), yet changes in groundwater storage can lead to changes in hydraulic properties, which may be irreversible depending on geomechanical conditions (Galloway & Burbey, 2011, not my paper, just a reference). I recommend clarifying the assumptions under which the groundwater system is considered reversible in your simulations or rephrasing the sentence to avoid misunderstanding.

We agree with reviewer#2 comment. In our original comment, we were referring to projected head changes without considering compaction effects and associated storativity reductions. In Galloway and Burbey (2011), the driver of regional land subsidence is aquifer overexploitation. Such a process in the study area could potentially be triggered due to drainage of peatlands, dewatering of open-pit mines, and extensive pumping in the urban area (Wolkersdorfer and Thiem, 1999; Landgraf, 2022).

That being said, we are not aware of any studies of climate change directly contributing to the subsidence due to reduction of groundwater recharge. We propose to revise the statement as follows:

Our results also indicate that the tested magnitude of projected recharge changes (-10 to 20 %) only influences groundwater dynamics (e.g., hydraulic gradients and flux) to a limited extent, as compared with stronger changes due to pumping. More fundamental changes in basin-scale flow due to climatic variability

would require sustained forcing beyond the 2100 horizon and evaluation through thermo-hydraulic-mechanical simulations.

**References:**

Brunner, L., Pendergrass, A. G., Lehner, F., Merrifield, A. L., Lorenz, R., and Knutti, R.: Reduced global warming from CMIP6 projections when weighting models by performance and independence, Earth Syst. Dynam., 11, 995-1012, 10.5194/esd-11-995-2020, 2020.

Eyring, V., Bony, S., Meehl, G. A., Senior, C. A., Stevens, B., Stouffer, R. J., and Taylor, K. E.: Overview of the Coupled Model Intercomparison Project Phase 6 (CMIP6) experimental design and organization, Geosci. Model Dev., 9, 1937-1958, 10.5194/gmd-9-1937-2016, 2016.

Frick, M., Scheck-Wenderoth, M., Schneider, M., and Cacace, M.: Surface to Groundwater Interactions beneath the City of Berlin: Results from 3D Models, Geofluids, 2019, 4129016, <a href="https://doi.org/10.1155/2019/4129016">https://doi.org/10.1155/2019/4129016</a>, 2019.

Galloway, D. L. and Burbey, T. J.: Review: Regional land subsidence accompanying groundwater extraction, Hydrogeology Journal, 19, 1459-1486, 10.1007/s10040-011-0775-5, 2011.

Landgraf, L.: Das Moorschutzfachkonzept Brandenburgs—wie gelingt der Klimaschutz auf Moorböden in der Praxis?, TELMA-Berichte der Deutschen Gesellschaft für Moor-und Torfkunde, 52, 129-154, 2022.

LfU: Landesamt für Umwelt Brandenburg - Auskunftsplattform Wasser (APW), <a href="https://apw.brandenburg.de/">https://apw.brandenburg.de/</a>, last access: 01/10/2024.

Merrifield, A. L., Brunner, L., Lorenz, R., Humphrey, V., and Knutti, R.: Climate model Selection by Independence, Performance, and Spread (ClimSIPS v1.0.1) for regional applications, Geosci. Model Dev., 16, 4715-4747, 10.5194/gmd-16-4715-2023, 2023.

Nguyen, V. D., Vorogushyn, S., Nissen, K., Brunner, L., and Merz, B.: A non-stationary climate-informed weather generator for assessing future flood risks, Adv. Stat. Clim. Meteorol. Oceanogr., 10, 195-216, <a href="https://doi.org/10.5194/ascmo-10-195-2024">https://doi.org/10.5194/ascmo-10-195-2024</a>, 2024.

Noack, V., Scheck-Wenderoth, M., Cacace, M., and Schneider, M.: Influence of fluid flow on the regional thermal field: results from 3D numerical modelling for the area of Brandenburg (North German Basin), Environmental earth sciences, 70, 3523-3544, https://doi.org/10.1007/s12665-013-2438-4, 2013.

Wolkersdorfer, C. and Thiem, G.: Ground water withdrawal and land subsidence in northeastern Saxony (Germany), Mine Water and the Environment, 18, 81-92, 1999.

---

## Author Comment (AC3)

The study by Tsypin et al. aims to quantify future groundwater warming under various climate change scenarios. Its key novelty lies in the explicit consideration of heat advection by groundwater flow, an aspect often overlooked in previous research.

The manuscript is well written and presents a compelling approach. Several comments should be addressed to further strengthen its impact.

We would like to thank reviewer#3 for their feedback on our study, as well as for constructive suggestions aimed to improving the manuscript's clarity. In the following, we provide our point-by-point responses to each comment (our answers are marked in red).

General:

- It would be beneficial for the reader to make titles of different chapters more detailed.

We reviewed the structure and chapter titles in the original manuscript. As reviewer#3 did not provide specific suggestions, we made minor adjustments based on our own judgment:

*4. Scenario Definition* to *4. Climate scenario definition*

*6.3 Effects of recharge projections on the thermal field* to *6.3 Effects of recharge projections on the subsurface thermal filed*

- Figures and legends need more clarity.

We have adjusted the figures mentioned by reviewer#3 and implemented additional improvements based on suggestions from the other reviewers.

Specific comments:

L57 Regarding the impact of GW dynamic on subsurface temperatures, you may also check Klepikova et al., ERL, 2025.

Yes, we are familiar with the findings of Klepikova et al. (2025), who described thermal effects arising from shifts in groundwater dynamics caused by groundwater abstraction. While the focus of our paper is on climate-driven changes in advection rates, we agree that studies by Klepikova et al. (2025) as well as Bense et al. (2020) are highly compelling, as multi-year pumping exert a stronger influence on groundwater fluxes than comparatively slow climate-driven recharge reduction. This study will be cited in the revised manuscript.

L67 As for the impact of advection on temperature profiles, a review by Kurylyk, 2018 should be cited.

Thank you for the suggestion. We will add this reference in the introduction, in addition to already cited older works.

L104 "representative of EU geology" sounds weird.

We re-phrased this sentence as follows: *The study area spans Germany's federal state of Brandenburg and the city of Berlin and exhibits climatic and geological conditions typical of north-central Europe.*

Fig.1 please provide scale reference on all subfigures.

Thank you for pointing this out. The corrected figure is shown below:

[Figure]

Section 3: In order to improve the clarity I believe that general conceptual models should be described first.

The Methods section organized according to the order in which the components of the integrated workflow were applied:

3.1 Simulation of downscaled weather time-series with the non-stationary Regional Weather Generator

3.2 Recharge estimation with a mesoscale Hydrologic model

3.3 Groundwater flow and transport modeling with the finite element method.

[Figure]

We are not entirely sure which "general conceptual models" reviewer#3 is referring to. If this comment is related to the advective-conductive heat transport in porous media, we find its description in Section 3.3 to be appropriate.

It could be argued that, given the objective of the study, the subsurface component should be introduced first, while the modeling of recharge and temperature time series, serving as boundary conditions for the groundwater model, could be described afterwards. However, we still prefer the current sequential structure of the Chapter and find it most appropriate for HESS journal, which targets a broad hydrology-oriented community.

L167-168 The main difference in between these scenarios should be already mentioned here.

We believe that the main difference between the scenarios is already described in the text: "*SSP245, a moderate-emission pathway where mitigation efforts balance sustainability measures with economic growth, and SSP585, a high-end emission scenario driven by fossil fuel development with minimal mitigation (IPCC, 2023)*". One consequence that should have been stated here more explicitly is that estimated global warming levels under SSP585 are substantially higher than under SSP245. More detailed quantitative differences between the two scenarios are region- and model- specific, and therefore discussed in detail in a dedicated chapter (Chapter 4).

L254 D of fluid?

D is the bulk thermal diffusivity, which in our model accounts only for heat conduction, without including mechanical dispersion:

$D = k/\rho c_p$,

where $k$ is thermal conductivity, $\rho$ is density, and $C_p$ is specific heat capacity. All three parameters are bulk properties, calculated as porosity-weighted average of the fluid and solid phases.

That being said, following a comment from reviewer#1, we decided to exclude the interpretation of *Pe* from the revised manuscript.

Figure 3. Expand and detail the legend.

A corrected figure with the expanded legend is provided below:

[Figure]

Figure 4. The clarity of this figure could be improved as well.

An updated version of the figure is provided below.

[Figure]

L343 Could you tell more about the T data (depth/profile or point?/range/precision)?

Temperatures from deep aquifers (>500 m) that we used to validate the initial thermal field, are a mix of (a) bottom-hole temperatures (BHT) obtained in wells shortly after drilling, thus representing perturbed borehole temperatures and (b) continuous temperature logs, mostly measured after the boreholes were cased (Förster, 2001). All BHT temperatures were previously corrected by Horner-plot correction and the exponential integral method based on a model simulating the temperature build-up during shut-in time of a well (Förster, 2001). The estimated error of the corrected measurements was ±3°C or ±10°C. Some temperature logs were also corrected using a simple empirical correction, with the error in final estimates of ±3°C (Förster, 2001).

Shallow groundwater temperature monitoring is available from a handful of wells in Berlin and is limited to 80 m below ground (SenStadt, 2020). Unfortunately, continuous vertical profiles and Distributed Temperature Sensing (DTS) are only available for the urban area. We chose to present them separately (Appendix B), due to strong land cover overprint.

L370 Please explain "teleconnections".

Our original sentence was imprecise and can be re-formulated without referring to teleconnections:

"Such behavior has been frequently observed in historical data and attributed to impact of intra-annual climatic fluctuations, such as North Atlantic Oscillation (Liesch and Wunsch, 2019)."

Figure 13. It is not really clear what do you want to highlight by horizontal arrows.

The black horizontal arrows delineate surface and geologic domains in the schematic cross-section. These domains have distinctions in shallow and deep groundwater dynamics. We agree that their recognition may be subjective. Therefore, in the figure caption we point reader to the detailed description of the domains in the main text. Slightly adjusted figure is presented below:

[Figure]

L535 This is confusing as the depth of penetration should be regulated by the GW flow rate and thermal properties of rocks.

We agree with reviewer#3 that the depth of warming penetration and the magnitude of temperature change at a given depth are regulated by flow velocities and thermal properties of rocks. However, they are also controlled by the magnitude of the applied surface forcing. For example, Kurylyk et al. (2015)

states: "*The thermal sensitivity formulae suggest that shallow groundwater will warm in response to climate change and other surface perturbations, but the timing and magnitude of the subsurface warming depends on the rate of surface warming, subsurface thermal properties, bulk aquifer depth, and groundwater velocity.*"

Our results indicate that, given the projected changes in surface temperature and the recharge rates, surface warming exerts a greater effect on the deviation of the geothermal gradient than shifts in recharge flux. To further support this, we present below the results from the analytical solution of Kurylyk et al. (2015) using the same degree of surface warming and recharge variability as applied in our numerical model.

[Figure]

L541-544 Rephrase this sentence to make it clearer (i.e. what is meant by "this response"?)

We rephrased this part as follows:

*The contrasting sensitivity of surface domains to climate change supports the conclusions of Burns et al. (2017) and Taniguchi (2021) that the thermal response of groundwater to surface warming is amplified in topographically elevated recharge areas (SD1) and attenuated in low-lying discharge areas (SD3). Such spatial differentiation in the magnitude of subsurface warming is controlled by the downward and upward components of the regional groundwater flow field, respectively.*

L544 differentiation of what?

See our response to the previous question.

L645 for which depth?

We thank reviewer#3 for pointing out this ambiguity. The corrected sentence now reads as:

*The magnitude of groundwater temperature increase depends primarily on the global warming levels: the mean sensitivity to surface temperature scenario is approximately 1.4 °C **at the water table**, while the difference between recharge scenarios contributes up to 0.4 °C between the present (2002-2021) and late-century (2081-2100) periods.*

References:

Bense, V. F., Kurylyk, B. L., de Bruin, J. G. H., and Visser, P.: Repeated Subsurface Thermal Profiling to Reveal Temporal Variability in Deep Groundwater Flow Conditions, Water Resources Research, 56, e2019WR026913, https://doi.org/10.1029/2019WR026913, 2020.

Förster, A.: Analysis of borehole temperature data in the Northeast German Basin: continuous logs versus bottom-hole temperatures, Petroleum Geoscience, 7, 241-254, 10.1144/petgeo.7.3.241, 2001.

Klepikova, M., Bense, V., Le Borgne, T., Guihéneuf, N., and Bour, O.: Impact of groundwater extraction on subsurface thermal regimes, Environmental Research Letters, 20, 054048, 10.1088/1748-9326/adc8bb, 2025.

Kurylyk, B. L., MacQuarrie, K. T. B., Caissie, D., and McKenzie, J. M.: Shallow groundwater thermal sensitivity to climate change and land cover disturbances: derivation of analytical expressions and implications for stream temperature modeling, Hydrol. Earth Syst. Sci., 19, 2469-2489, https://doi.org/10.5194/hess-19-2469-2015, 2015.

Liesch, T. and Wunsch, A.: Aquifer responses to long-term climatic periodicities, Journal of Hydrology, 572, 226-242, https://doi.org/10.1016/j.jhydrol.2019.02.060, 2019.

SenStadt: Senatsverwaltung für Stadt-entwicklung, Bauen und Wohnen - Berlin Environmental Atlas, Berlin, https://www.berlin.de/umweltatlas/en/water/groundwater-temperature/, 2020.